# Predicting clinical benefit of immunotherapy by antigenic or functional mutations affecting tumour immunogenicity

Kwoneel Kim[1,2,3,7], Hong Sook Kim[4,7], Jeong Yeon Kim[2], Hyunchul Jung [2], Jong-Mu Sun[4], Jin Seok Ahn[4], Myung-Ju Ahn[4], Keunchil Park[4], Se-Hoon Lee[4,5✉] & Jung Kyoon Choi[2,6✉]

Neoantigen burden is regarded as a fundamental determinant of response to immunotherapy. However, its predictive value remains in question because some tumours with high neoantigen load show resistance. Here, we investigate our patient cohort together with a public cohort by our algorithms for the modelling of peptide-MHC binding and inter-cohort genomic prediction of therapeutic resistance. We first attempt to predict MHC-binding peptides at high accuracy with convolutional neural networks. Our prediction outperforms previous methods in > 70% of test cases. We then develop a classifier that can predict resistance from functional mutations. The predictive genes are involved in immune response and EGFR signalling, whereas their mutation patterns reflect positive selection. When integrated with our neoantigen profiling, these anti-immunogenic mutations reveal higher predictive power than known resistance factors. Our results suggest that the clinical benefit of immunotherapy can be determined by neoantigens that induce immunity and functional mutations that facilitate immune evasion.

[1] Department of Biology, Kyung Hee University, Seoul 02447, Republic of Korea. [2] Department of Bio and Brain Engineering, KAIST, Daejeon 34141, Republic of Korea. [3] Clinical Research Center, Asan Institute for Life Sciences, Asan Medical Center, Seoul 138736, Republic of Korea. [4] Division of Hematology/Oncology, Department of Medicine, Samsung Medical Center, Seoul 06351, Republic of Korea. [5] Department of Health Sciences and Technology, Samsung Advanced Institute of Health Science and Technology, Sungkyunkwan University, Seoul 06351, Republic of Korea. [6] Penta Medix Co., Ltd., Seongnam-si, Gyeongi-do 13449, Republic of Korea. [7] These authors contributed equally: Kwoneel Kim, Hong Sook Kim. ✉email: shlee119@skku.edu; jungkyoon@kaist.ac.kr

Cancer immunotherapy has become remarkably effective in a range of human cancers. In particular, checkpoint blockade therapies, such as anti-CTLA-4 (ipilimumab) and anti-PD-1 (nivolumab and pembrolizumab), are able to reverse tumour-induced immunosuppression and induce durable clinical responses[1]. Tumour cells produce neoantigens or antigens that the immune system never encountered without cancer. The epitopes of neoantigens, which are displayed with major histocompatibility complexes (MHCs) on the surface of cancer cells, provoke immune response when recognised by T cells. Tumours loaded with more neoantigens therefore are more likely to be responsive to the anti-immunosuppressive strategies[2]. However, a sizeable fraction of those tumours resist to checkpoint blockade[3].

Tumour neoantigens are generated by somatic mutations producing novel peptides that can be recognised as foreign, thereby conferring immunogenicity to cancer cells. Neoantigen burden is therefore regarded as a fundamental determinant of response to immunotherapy including checkpoint blockade. Neoantigen burden has been estimated by several computational tools that predict peptide binding to MHC class I[4]. However, the current tools fail to capture the nonlinear high-order features of interactions among different amino acids[5]. In this regard, a recent study has shown that amino acids distal to contact interfaces can exert significant effects on the interactions between MHC-peptide complexes and T cell receptors (TCRs)[6]. Therefore, we are in need of a prediction method that captures the spatial features amino acid interactions. Convolutional neural networks (CNNs) have been applied successfully for the identification of local sequence patterns in protein-nucleic acid interactions[7] or functional effects of noncoding variants[8]. Hence, in this work, we attempted to develop a CNN-based algorithmic framework for the prediction of peptide binding to MHC class I molecules.

There are additional putative biomarkers that have been reported to predict response to checkpoint blockade together with neoantigen burden. Tumour heterogeneity[9], copy number alteration[10], aneuploidy[11], and genetic alteration of specific genes[12] or pathways[13] have been identified as resistance markers. Pre-existing T cell infiltration also may impinge on response to checkpoint blockade[14,15]. Since all of these markers or factors differ greatly between individual cancer patients, a framework for describing diverse predictive factors has been proposed to enable personalised cancer immunotherapy[16]. However, resistance mechanisms involving alterations of individual genes have not been explored at the genomic level. Especially, tumours loaded with neoantigens should carry a large number of functional mutations due to high mutation rates. The functional mutations that facilitate immune evasion may be subject to positive selection. In this work, we examine whether mutation profiles can explain the therapeutic resistance of tumours loaded with neoantigens. We first use our clinical data together with public cohort data and then perform further tests using molecular immune evasion markers in other tumour types.

## Results

**Accurate prediction of peptide-MHC class I binding**. We constructed a convolutional neural network (CNN) model to predict the binding of MHC class I molecules and peptides. About 50,000 binding data obtained from the immune epitope database (IEDB) 3.0. (http://www.iedb.org/)[17] were used for the model training. The CNN architecture enables to capture specific local properties of input data such as in images. To build an amino acid interaction map for our CNN model, we inferred the binding preference of each pair of amino acids using interaction energy estimated from the frequency of neighbouring amino acids in native protein structures[18]. The interaction map between

**Table 1 Information of cohorts used in this work.**

| Cohort name | Tumour type | Cohort size | Target checkpoint | Reference |
|---|---|---|---|---|
| SMC | Lung cancer | 122 | PD-1/PD-L1 | This work |
| Hellmann | Lung cancer | 75 | PD-1 & CTLA-4 | Ref. 25 |
| Rizvi | Lung cancer | 34 | PD-1 | Ref. 24 |
| Van Allen | Melanoma | 110 | CTLA-4 | Ref. 21 |
| Snyder | Melanoma | 64 | CTLA-4 | Ref. 22 |
| Roh | Melanoma | 56 | PD-1 & CTLA-4 | Ref. 10 |
| Riaz | Melanoma | 68 | PD-1 | Ref. 12 |

peptides and HLA sequences were scanned by a number of kernels to detect particular binding motifs that are critical for peptide-MHC I binding (Fig. 1a and Supplementary Fig. 1). According to the receiver operating characteristic (ROC) curves, the area under the curve (AUC) for the training data was 0.93 for HLA-A and 0.94 for HLA-B, respectively. In the test data, the AUC was 0.89 for HLA-A and 0.86 for HLA-B.

Comparison with an interaction map consisting of randomly permuted or null values made it clear that the amino acid interaction map we used for our model played a critical role in predicting the MHC-peptide binding (Supplementary Fig. 2). This implies that the amino acid interaction preferences derived from native protein structures[18] can serve as interaction parameters for the modelling of the binding of MHCs and peptides. We also tested whether the mere sum of the amino acid binding preferences in the interaction map can be used as a predictor. If this is the case, the sum of the preferences should be lowest for true positives because the binding preferences were encoded as the energy levels of interactions. However, we could rule out this possibility (Supplementary Fig. 4), indicating that our CNN model was trained on the pattern rather than the aggregate of binding preferences.

Next, we tested the prediction accuracy of our CNN model using the IEDB test datasets that are weekly updated along with performance evaluations of current prediction tools including the most widely used one named NetMHCpan[19] (now updated to NetMHCpan 4.0[20]). Our model showed higher performance than the compared prediction tools. In terms of AUC, our method was superior to SMMPMBEC, ANN, NetMHCcon, and NetMHCpan for 100%, 100%, 90%, and 70% of the test cases. With regards to the F1 score, our CNN was superior to all the methods for 80% of the test cases (Fig. 1b). Highest performance was achieved for binding of HLA-As and 9-mer peptides. In general, the training dataset was largest for this class of binding, implying that higher prediction accuracy can be expected as in vitro binding data grow. In contrast to the size of training data, prediction accuracy was maintained irrespective of the size of test data (Supplementary Fig. 3).

**Clinical and molecular relevance of neoantigen load estimated by CNN**. We applied our CNN model to predict MHC class I-binding neoantigens for clinical samples that were used in four cohorts of melanoma[10,21–23] and three cohorts of lung cancer[24,25], including our clinical cohort referred to as SMC (Samsung Medical Center) cohort (Table 1 and Supplementary Data 1). The samples were divided by curated clinical response, and the extent of neoantigen load was assessed on the basis of our prediction model. Neoantigen load predicted by the CNN method showed a significant correlation with clinical benefit in most cohorts (Fig. 2a). In particular, the P values of our association were significant except two cohorts, for which marginal or lower

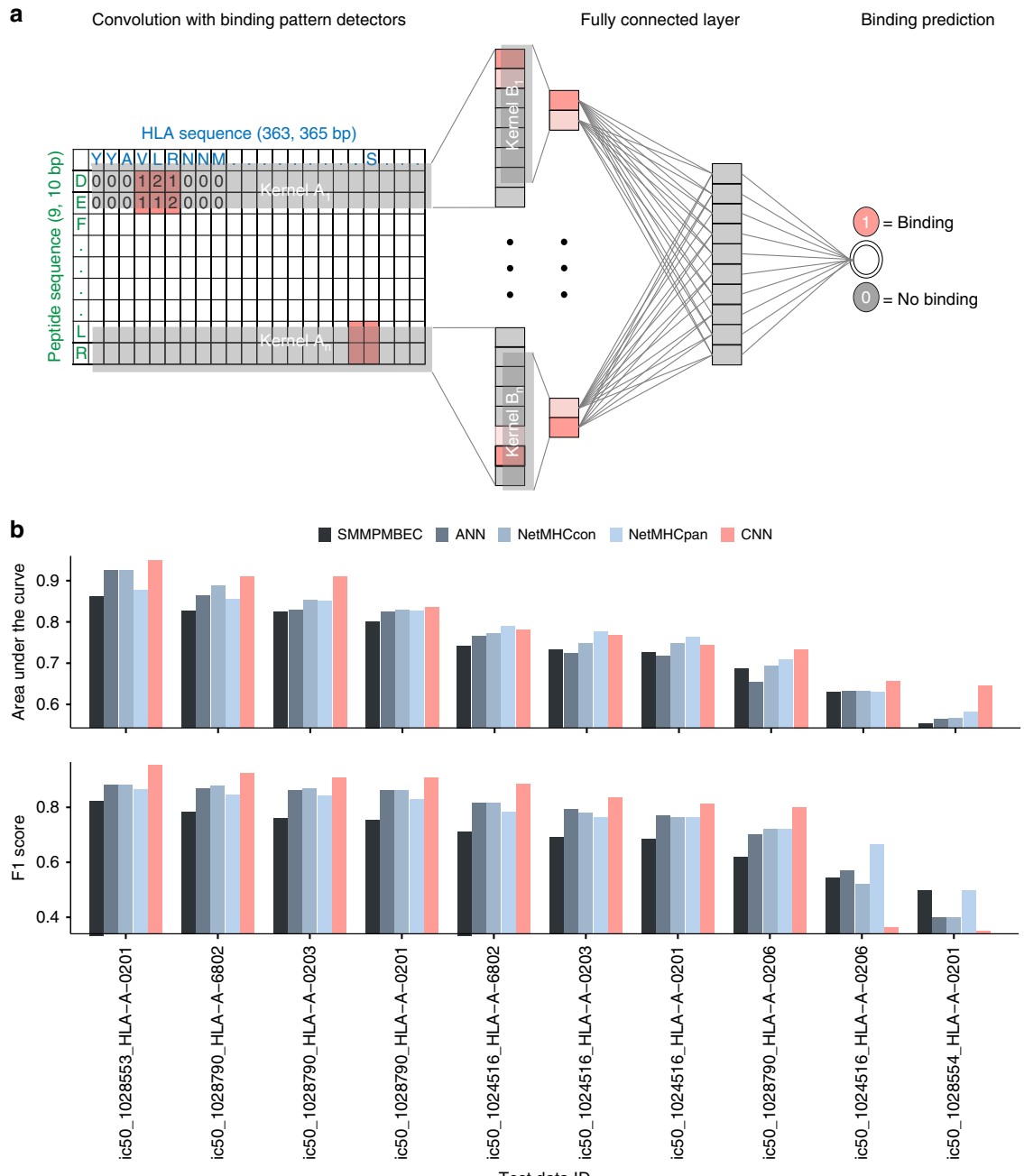

**Fig. 1 Prediction model for peptide-MHC class I binding and performance evaluation. a** CNN architecture was used to predict binding between peptides and MHC class I molecules. The two-dimensional map of interactions between amino acids in peptide-MHC class I complex was used as the input matrix. A set of kernels, $A_1,...A_n$, covering the entire HLA sequence were applied on the input matrix. The output convolution scores in the first layer were scanned by the second set of kernels, $B_1,..., B_n$. A fully connected layer attached to the second layer integrated the convoluted patterns for classification. **b** Comparison of prediction performance with SMMPMBEC, artificial neural network (ANN), NetMHCcon, and NetMHCpan on the basis of weekly updated test datasets of IEDB. In terms of AUC, our method was superior to SMMPMBEC, ANN, NetMHCcon, and NetMHCpan for 100%, 100%, 90%, and 70% of the test cases. With regards to the F1 score, our CNN was superior to all the methods for 80% of the test cases. Source data are provided as a Source Data file.

associations were observed when NetMHCpan was used for neoantigen prediction. We compared patient survival between the high- and low-neoantigen groups for the cohorts showing the significant associations. Higher neoantigen burden was significantly associated with longer disease free time consistently in all cohorts (Fig. 2b). This correlation was not seen when NetMHCpan was used (Supplementary Fig. 5).

We wanted to further validate the relevance of the CNN predictions using samples from The Cancer Genome Atlas

(TCGA) although not in the setting of checkpoint blockade therapy. To this end, we collected exome and transcriptome data for skin cutaneous melanoma, performed HLA typing for MHC class I, and then predicted neoantigens that were capable of binding the relevant MHC I molecules by using our CNN. We also computed the immune score and TCR diversity for the same samples. The diversity of tumour-reactive T cell clonotypes represented by TCR repertoire can reflect the load of immunogenic antigens that stimulate T cell infiltration[26]. Similarly, the

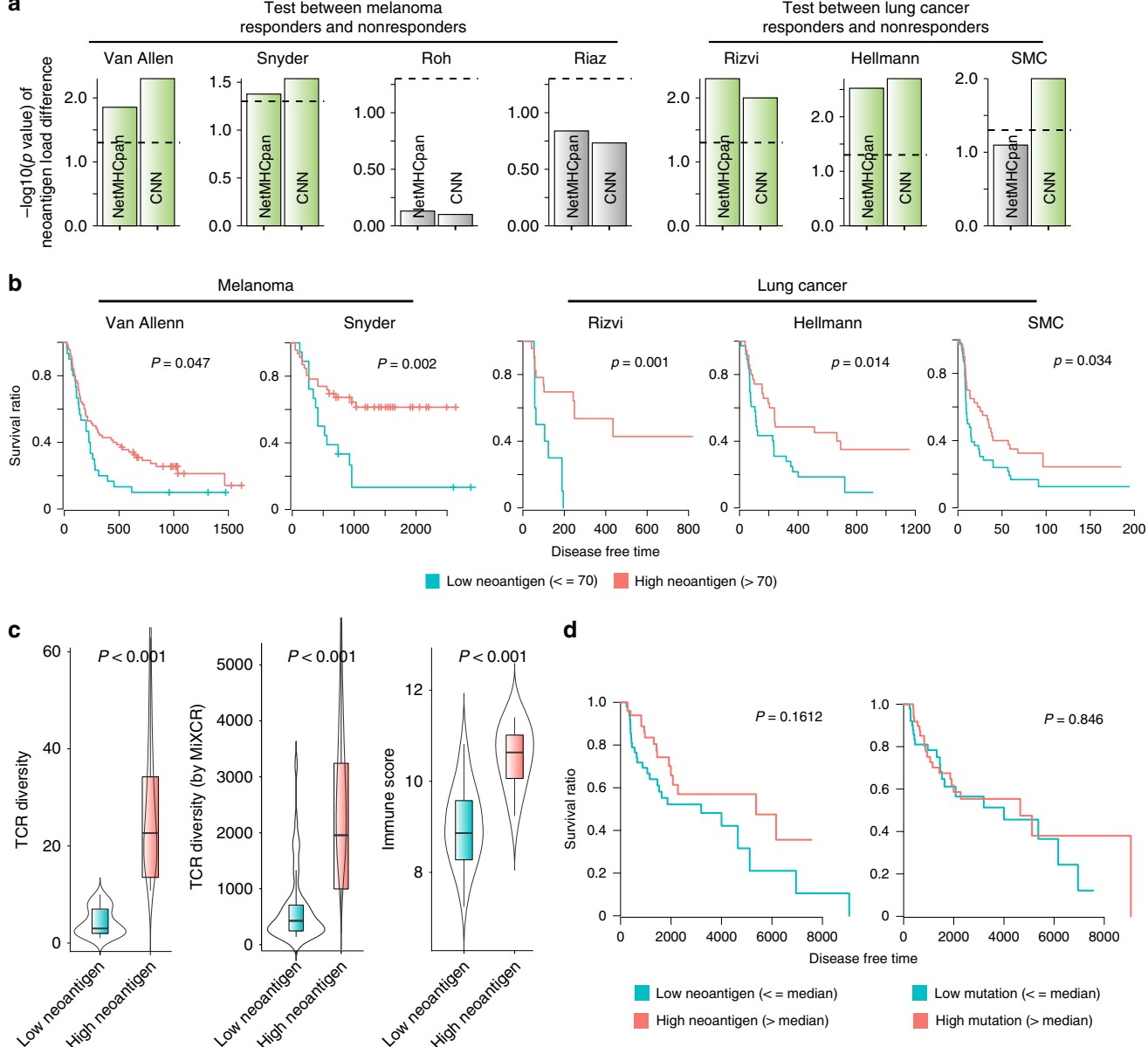

**Fig. 2 Clinical relevance of neoantigen load predicted by the CNN model. a** Neoantigen load was estimated by the CNN and NetMHCpan method for four melanoma cohort (Van allen, Snyder, Roh, and Riaz) and three lung cancer cohort (Rizvi, Hellmann, and SMC) samples divided according to the clinical benefit to checkpoint blockade. The Wilcoxon rank-sum test was used to calculate the statistical significance of the difference in neoantigen load between the two groups. -log10(P value) was plotted with the significant cases highlighted in green. **b** Survival analysis was performed for samples with high versus low neoantigen load in the two melanoma and three lung cancer cohorts that exceeded a given threshold in **a**. The same threshold of neoantigen load as the previous study[9] was used. **c, d** To test clinical relevance on TCGA melanoma samples (SKCM), we computed TCR diversity and immune score for SKCM and compared them between high-neoantigen ($n = 52$ for TCR diversity, $n = 52$ for TCR diversity by MiXCR, and $n = 11$ for Immune score, respectively) and low-neoantigen groups ($n = 51$ for TCR diversity, $n = 51$ for TCR diversity by MiXCR, and $n = 10$ for Immune score, respectively). The centre line and bottom/upper bounds indicate the median and 1st/3rd quartile, respectively. **c** Also, survival analysis was performed for samples with high ($n = 52$) versus low ($n = 51$) neoantigen load (**d**). The median level of neoantigen load was used as the threshold to divide the neoantigen groups. Source data are provided as a Source Data file.

immune score, defined as the geometric mean of the expression levels of immunologically relevant genes, has been reported to correlate with the clinical benefit of checkpoint blockade immunotherapy[10]. According to our analyses, neoantigen load estimated by the CNN model showed a significant association with the immune score and TCR diversity (Fig. 2c). For these samples, our estimate of neoantigen load (left of Fig. 2d) than mutation burden (right of Fig. 2d) showed a better correlation with patient survival. However, this should be interpreted with caution given no statistically significance, which may be attributed to the fact that we simply compared patient survival but not in the setting of checkpoint therapy.

**Functional immune mutations explain poor therapeutic response**. A large amount of neoantigens reflects a high mutation frequency. If some of protein-changing mutations perturb immune reaction, tumours carrying these mutations will not be responsive to immunotherapy despite high neoantigen load.

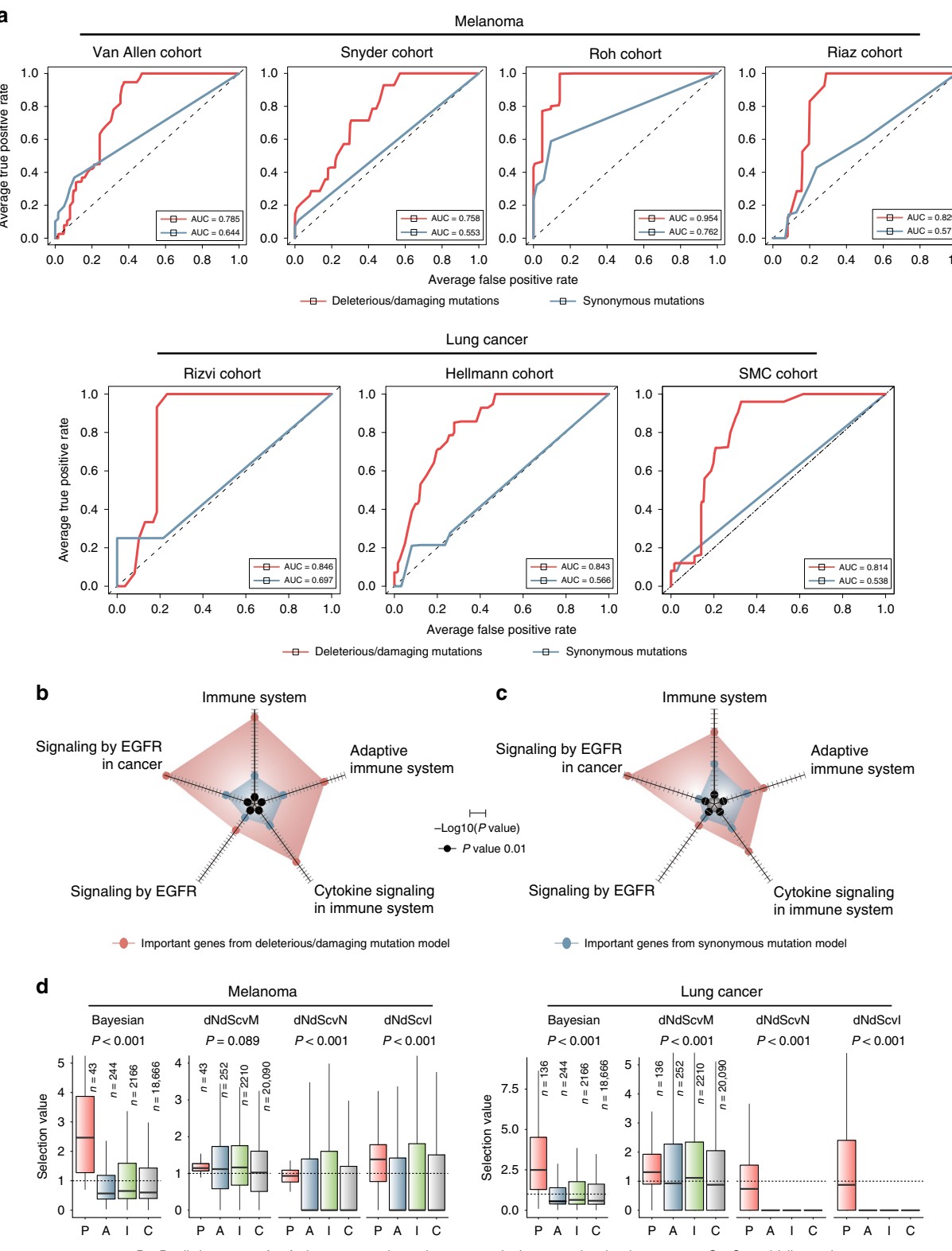

P = Predictive genes,  A = Antigen-presentation pathway genes,  I = Immune-related pathway genes,  C = Control (all genes)

However, there was no statistical significance when we directly compared the responsive and resistant groups in mutation burden on genes involved in the immune system, immune response, antigen presentation, or antigen processing (Supplementary Fig. 6). Mutation burden was generally higher in the resistant tumours for genes involved in antigen presentation or processing but only marginally (Supplementary Fig. 6). This suggests a role

for genes that can affect immune response indirectly, for example, by interacting with genes that are formally annotated as immune related.

We therefore developed a classifier that uses the exomic profiles of functional mutations to predict samples that will be resistant to checkpoint blockade. Our MHC-binding prediction was performed for all the cohorts (Table 1) to select samples with

**Fig. 3 Exomic prediction of therapeutic resistance. a** We trained random forests using genes that harbour deleterious or damaging mutations in > 5% of the samples. For each cohort, we used the other cohorts of the same tumour type as training data. For example, we trained random forests with Hellman and Rizvi cohorts and tested performance on SMC cohort. Shown here are the ROC curves comparing the original data (red curves) and negative controls generated by training the classifier on synonymous mutations (blue curves). The same number of features and samples were used between the original and negative control model. **b**, **c** Functional enrichment of genes with high explanatory power (variable importance > 3) and their interaction partners in **b** melanoma and **c** lung cancer. The radar plots present the statistical significance of enrichment. The axis length scales with -$\log_{10}$(P value). **d** Selection values based on the Bayesian inference[32] and covariate model (*dNdScv*)[33] for the genes with high variable importance from our random forest classifier. Shown are the selection values obtained for skin cutaneous melanoma (SKCM) and lung squamous cell carcinoma (LUSC) samples from TCGA. dNdScvM and dNdScvN are the normalised ratio of nonsynonymous to synonymous mutations (dN/dS) for missense and nonsense mutations, respectively. dNdScvI indicates the observed to expected ratio for indels. The centre line and bottom/upper bounds indicate the median and 1st/3rd quartile, respectively. Source data are provided as a Source Data file.

high neoantigen burden and define the resistant group according to the clinical outcome (Supplementary Fig. 7). For each cohort, we used the other cohorts of the same tumour type as training data. For example, we trained random forests with Hellman and Rizvi cohorts and tested performance on SMC cohort. As a negative control, we trained the classifier on synonymous mutations. As a result, only the models trained with functional mutations achieved reasonable accuracies for all the test datasets (Fig. 3a).

We next investigated which genes were critical for predicting therapeutic response when mutated. For this, we retrieved genes with a high 'variable importance' of random forests and their interacting partners in the protein interactome[27–29] (Supplementary Data 2 and 3). These genes were related to the adaptive immune system, cytokine signalling, and epidermal growth factor receptor (EGFR) signalling (Fig. 3b, c). Tumours produce cytokines that alter tumour immunogenicity and the antitumour immune response of the host. For example, interferon-γ pathway inactivation in tumours was implicated in resistance to immunotherapy[13]. The EGFR pathway was implicated in immune escape and clinical response to immunotherapy[30,31].

We then examined whether the predictive genes identified from our classifier carry the signatures of positive selection. Recent studies investigated selection patterns at the gene level based on the ratio of nonsynonymous to synonymous mutations across a large number of tumour samples[32,33]. Positive selection on mutations will lead to the excess of nonsynonymous mutations for given background mutation rates estimated by the frequency of synonymous mutations. In other words, a gene under positive selection will carry an extra complement of driver mutations in addition to passenger mutations. We employed the scores that were previously calculated for each gene by the Bayesian inference[32] and statistical model for covariates (*dNdScv*)[33] based on the mutation patterns observed in the TCGA data. Using these scores, we compared the degree of positive selection on the predictive genes with that on genes categorised as antigen-presentation or immune-related pathway. The score distributions for all genes were also considered. As a result, significantly higher positive selection scores, in particular those from the Bayesian inference[32], were observed for the predictive genes than for the other groups of genes (Fig. 3d), indicating that our prediction model was based on functional mutations that are subject to positive selection because of their contribution to immune evasion.

**Profiling functional mutations together with neoantigens accurately predicts therapeutic response.** Up to this point, we sought to resolve the contradiction between clinical observations and the "neoantigen roulette" theory[3] by a more accurate estimation of neoantigen load as a predictive marker of clinical response and by the identification of functional mutations in immune-related genes for prediction of therapeutic resistance.

We next wanted to evaluate the clinical utility of combining these two approaches on the basis of the melanoma samples in Roh cohort and lung cancer samples in SMC cohort that came with information on therapeutic response to checkpoint inhibitor and known resistance parameters such as copy number alteration (gain or loss) and tumour heterogeneity[9–11].

We performed regression of therapeutic resistance on the exomic prediction score from our classifier (Fig. 4a, d), on previously known resistance parameters (Fig. 4b, e), and on all the variables together (Fig. 4c, f). Overall, our classifier performed markedly better when tumours with high neoantigen load were defined by using our CNN than NetMHCpan (Fig. 4a–c for SMC cohort and Fig. 4d–f for Roh cohort). Most importantly, our exomic prediction score was the most significant contributing factor regardless of the regression methods in both cohorts (Fig. 4c, f).

**Functional immune mutations are associated with immune evasion signatures.** To test whether this approach is applicable to other cancer types, we used exome and transcriptome data from TCGA. For training of mutational patterns, we selected four tumour types for which a sufficient number of samples were available in the database: bladder cancer (BLCA, $n = 152$), oesophagus cancer (ESCA, $n = 36$), head and neck cancer (HNSC, $n = 242$), and lung cancer (LUSC, $n = 78$). We performed HLA typing and our CNN prediction to obtain the estimate of neoantigen load. Among the genes that are used for calculating the immune score, we chose those that showed the largest positive correlation with neoantigen load in each tumour type. Then, we identified the samples with immune evasion by looking for the cases that deviated from the correlation by showing a low expression level of those genes despite high neoantigen load. These samples can be regarded as the equivalent of the resistant tumours in the melanoma cohorts.

Using these data, we repeated the same analyses as we did with melanoma. For each tumour type, we trained random forest on the samples showing the immune evasion signatures by using as features the genes that harbour deleterious or damaging mutations in > 5% of all samples. The performance of our classifier was evaluated by comparing the functional mutation model with the control model trained with synonymous mutations (Fig. 5a). We next investigated which genes exerted explanatory power in predicting immune escape when mutated. In general, the functional mutation model resulted in genes with high variable importance (Fig. 5b). We retrieved these genes in each tumour type and collected their interacting partners in the protein interactome[27–29]. Similar to the melanoma cases, these genes tended to be involved in cytokine signalling pathway, adaptive immune system, and EGFR signalling in cancer (Fig. 5c and Supplementary Fig. 8).

We finally assessed the contribution of the mutation features in explaining the molecular signatures of immune evasion across

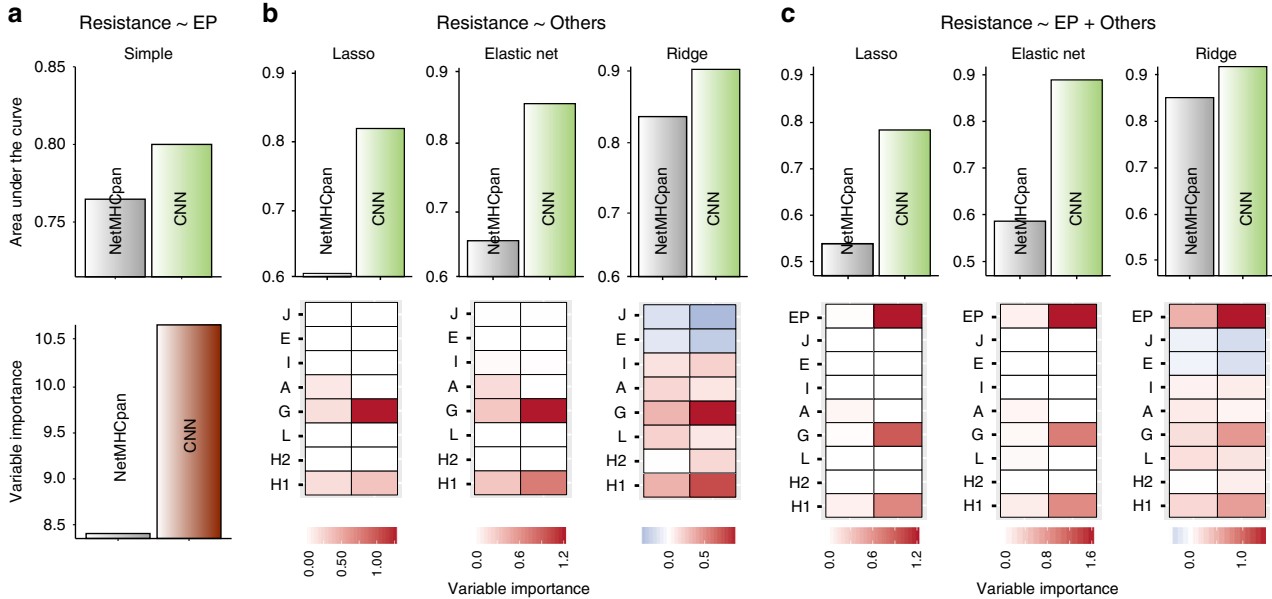

EP = Exomic prediction, J = JAK mutation, E = EGFR mutation, I = Immune-related pathway mutation, A = Antigen-presenting pathway mutation
G = Copy number gain, L = Copy number loss, H1 = Heterogeneity by expands, H2 = Heterogeneity by SciClone

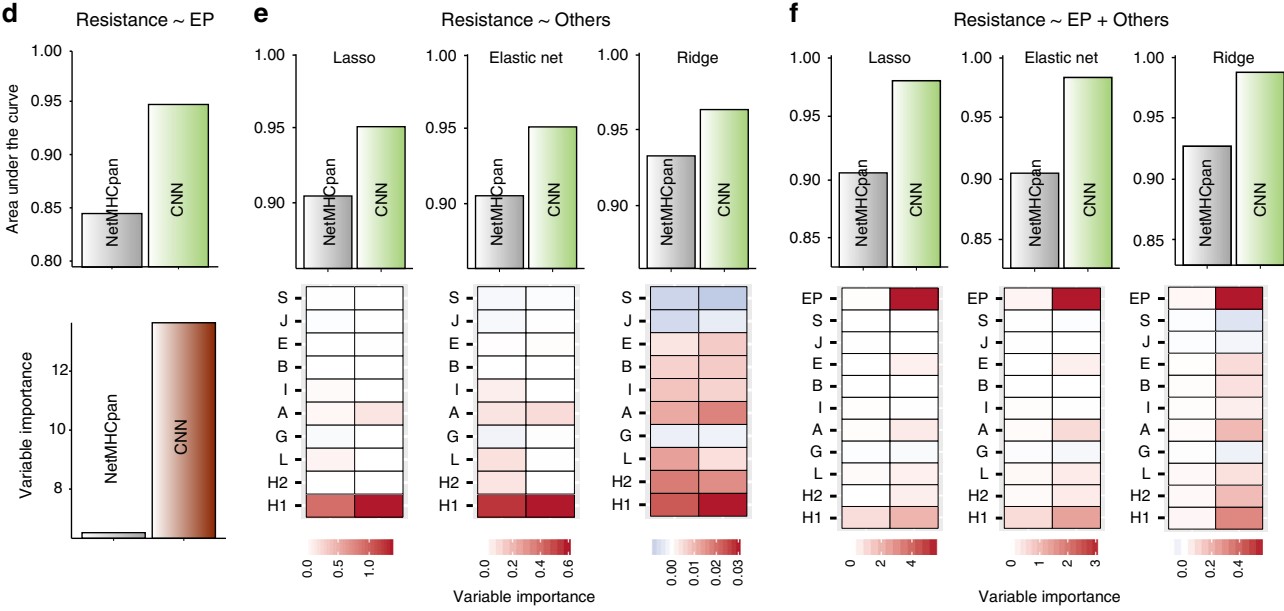

EP= Exomic prediction, S = SERPINB mutation, J = JAK mutation, E = EGFR mutation, B = B2M mutation, I = Immune-related pathway mutation,
A = Antigen-presenting pathway mutation, G = Copy number gain, L = Copy number loss, H1 = Heterogeneity by expands, H2 = Heterogeneity by SciClone

**Fig. 4 Comparison of exomic prediction with other resistance markers. a–c** Results for SMC cohort. **d–f** Results for Roh cohort. **a**, **d** AUC (upper) and variable importance (lower) for the regression of therapeutic resistance on the exomic prediction scores when resistance was defined with neoantigen load estimated by CNN or NetMHCpan. **b**, **e** AUC (upper) and variable importance (lower) for the regression of therapeutic resistance on the known resistance parameters when resistance was defined based on neoantigen load estimated by CNN or NetMHCpan. **c**, **f** AUC (upper) and variable importance (lower) for the regression of resistance on the known resistance parameters and exomic prediction scores when resistance was defined based on neoantigen load estimated by CNN or NetMHCpan. Source data are provided as a Source Data file.

different tumour types. There was a recent study that reported the correlation of extensive somatic copy number alterations (SCNAs) with the immune score[11]. While focal SCNAs mainly correlated with proliferation markers, arm- and chromosome-level SCNAs were negatively associated with the immune score. Multiple regression of immune evasion status on tumour heterogeneity, focal SCNA, arm/chromosome-level SCNA, and mutation-based prediction revealed the mutation features as the most significant contributor (Fig. 5d). Arm- and whole-

chromosome SCNAs generally stood out as the second important parameter in predicting immune evasion (Fig. 5d).

## Discussion

In this work, we first developed a prediction model for neoantigen identification. This model was trained on two-dimensional interaction patterns for peptide-MHC class I binding in contrast to one-dimensional sequential modelling of current

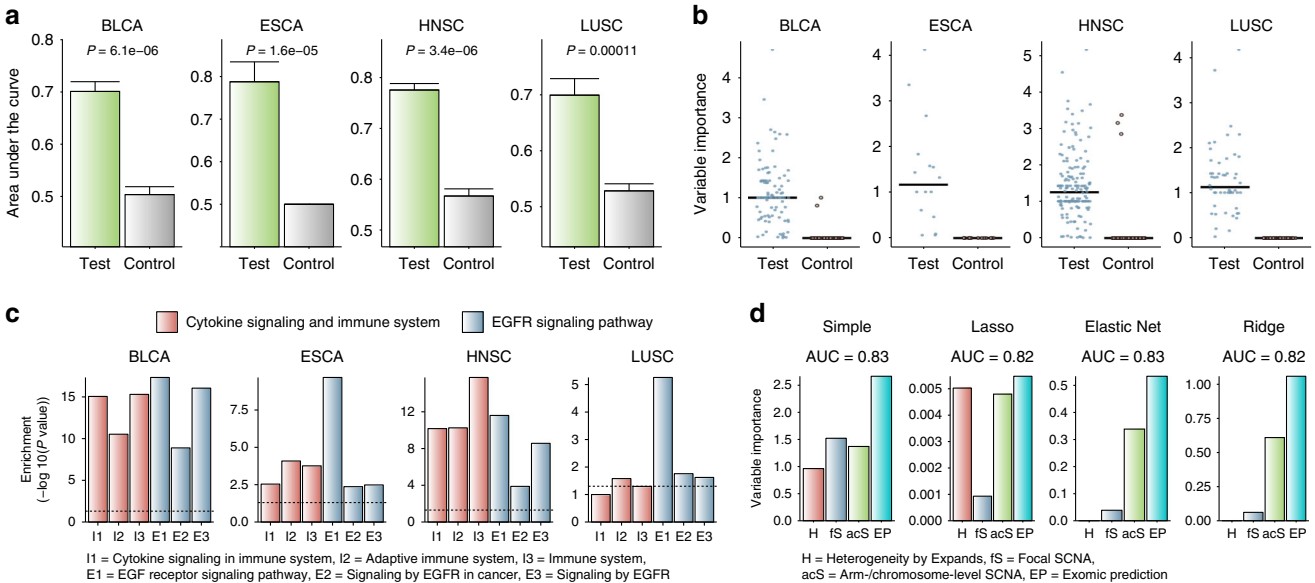

**Fig. 5 Immunogenic mutations as determinants of immune evasion of tumours. a** We trained random forest on TCGA samples of different tumour types using genes that harbour deleterious or damaging mutations. Tumours with immune evasion were defined by high neoantigen load and low activity of selected immune markers for each tumour type. Performance was evaluated using an independent test dataset that was separated from the training processes. Shown here are AUC values resulted from 100 repetitions that compared the original model (green) and negative control model (grey) generated by training the classifier on synonymous mutation patterns. In each repetition, the same number of features and samples were selected for the test and control model. The error bars indicate the standard error of the AUC values from the 100 classifiers. **b** Variable importance estimated by the test and control random forest models for the genes ($n = 99$ for BLCA, $n = 16$ for ESCA, $n = 160$ for HNSC, and $n = 52$ for LUSC, respectively) used as input features. **c** Functional enrichment of genes with high explanatory power and their interacting partners in the protein interactome. **d** Variable importance from the multiple regression of immune evasion status on neoantigen load and resistance parameters including the exomic prediction score. For the regression analyses, we obtained the mutation prediction scores from a leave-one-sample-out cross validation approach. Therefore, the prediction score was assigned to each sample from the classifier that did not include the given sample for training. Source data are provided as a Source Data file.

prediction tools. We then examined potential determinants of resistance to immunotherapy. While mutational patterns stood out, tumour heterogeneity also showed considerable explanatory power when our method was used for neoantigen prediction. A recent study presented the relevance of neoantigen heterogeneity to resistance of tumours with high mutation load[9]. Thus, tumour heterogeneity may confer resistance especially when neoantigen load is high.

Our methods focused on resistance of high-neoantigen tumours. However, there are samples that show sensitivity to checkpoint blockade despite a low estimate of neoantigen load. Hypersensitivity to the immune response can be a harmful trait for tumour growth. In contrast to mutations that promote immune evasion, such somatic variations that increase immune reaction will be eliminated by negative selection during tumour evolution. Therefore, this trait can be explained more likely by the genetic makeup of the donor rather than by somatic mutations. For example, a genetic polymorphism in the CTLA-4 gene was reported to alter the CTLA-4-driven regulation of T cell activation in the context of autoimmune disease[34]. A possible approach thus is to use the genotypes of the hypersensitive samples as predictive features for the classification of clinical response to immunotherapy. In this manner, the cases that contradict the neoantigen roulette[3] by showing resistance despite high neoantigen load or hypersensitivity despite low neoantigen levels could be predicted only based on their somatic or germline genetic profile. It needs to be emphasised that the success of this approach will hinge on the accurate estimation of neoantigen burden.

The recently proposed concept of "cancer immunogram"[16] included seven parameters that could characterise aspects of

cancer-immune interactions to determine treatment options in a more refined and personalised manner. Among those seven parameters, our study conferred immediate clinical utility to "tumour foreignness" and brought "tumour sensitivity to immune effectors" up to the systems level where potentially all available genes could be examined. There have been attempts to discover individual factors such as a particular gene[12] or pathway[13]. However, this is the first attempt to profile mutations at the whole-exome level. In doing so, we discovered that the EGFR pathway could be a primary target of somatic disruption for immune evasion. Our results indicate that a large fraction of variation in the clinical benefit of immunotherapy can be explained by contrasting effects of antigenic versus functional mutations on tumour immunogenicity. This approach offers practical advantage as well because calling single nucleotide variants is straightforward and less technically challenging than determining other parameters such as copy number alterations and tumour heterogeneity.

## Methods

**IEDB peptide-MHC binding data sets.** All training data for the prediction of peptide-MHC class I binding were obtained from IEDB 3.0. (http://www.iedb.org/)[17]. This database provided 57,173 data points consisting of binding affinity in terms of $IC_{50}/EC_{50}$ nM for 14,234 true (binding) and 42,879 false (non-binding) experiments. We used the affinity threshold that is commonly used to determine peptide-MHC binding to classify binding ($IC_{50}/EC_{50} < 500$ nM) and no binding ($IC_{50}/EC_{50} \geq 500$ nM). The major subset of the data, composed of two classes (HLA-A and HLA-B) of MHC class I against 9-mer and 10-mer peptides, was used for the development of our prediction method. To evaluate prediction performance, we employed weekly updated test datasets used for an automated benchmarking of selected peptide-MHC class I binding prediction tools[4]. This enabled us to compare the performance of our prediction model with the reported performance of the tools used for the benchmarking.

We used the weekly updated datasets that consisted of >10 binding affinity experiments.

**Peptide-MHC class I binding prediction**. We used CNN architecture to perform prediction of peptide-MHC class I binding. The schematic diagram of our model is shown in Fig. 1a. First, two-dimensional interaction maps for amino acid sequence pairs of peptides and MHC class I molecules were constructed as input matrices. We employed amino-acid interaction preferences computed based on contacts between Cα atoms or between any atoms in native protein structures[18]. For the calculation of the interaction map, the dataset of 1,654 proteins from the PISCES server[35] was curated, and structural information from the PDB[36] was obtained. The connectivity matrix based on the Cα-Cα distance was generated for each protein based on the distance cutoff of 6.5 Å between Cα-Cα atoms of amino acids with the exclusion of nearest neighbours along the sequence. Atom-atom contacts between two amino acid residues were also used such that residues i and j were considered to be in contact if any atom of the residue i is within a distance of 4.5 Å with any atom of the residue j. In this case, nearest neighbours (i ± 2) along the sequence are not considered. We also prepared interaction maps consisting of randomly permuted or null values. The interaction parameters based on the Cα-Cα contacts showed highest validation performance (Supplementary Fig. 2) and thus were used as input. Our CNN model consists of sequential alternating convolution layers that extract interaction features at different spatial scales, a fully connected layer that integrates information from a full-length sequence, and a sigmoid output layer that computes the probability for binding of the given peptide and MHC protein. Each layer of the CNN model executes a linear transformation of the output from the previous layer by multiplying a weight matrix, the output of which is subjected to a nonlinear transformation by ReLU activation as described below. The weight matrices are learned during training in the process of minimising predictive errors.

A variety of kernels were tested to achieve as high performance as possible. The two convolution layers of our model performed 2D convolution operation after optimisation of settings with 50 kernels for the first layer, 10 kernels for the second layer, 1000 batches, the stride size of 1, and the kernel size of 5 × 183 for each layer. All convolution outputs were transformed by a rectified linear activation function (ReLU) lifting negative values to 0. The first convolution layer was designed to detect binding patterns with a moving window with step size 1 on the interaction parameters of amino acid pairs. In the higher-level convolution layer, each convolution kernel served as a binding pattern detector over the output of the previous layer. More formally, the convolution layer computed

$$\text{convolution}(X)_{ik} = \text{ReLU}\left(\sum_{m=0}^{M-1}\sum_{n=0}^{N-1} W_{mn}^k X_{i+m,n}\right) \quad (1)$$

where $X$ is the input, $i$ is the index of the output position, and $k$ is the index of kernels. Each convolution kernel $W_k$ is an $M \times N$ weight matrix with $M$ being the window size and $N$ being the number of input channels. A pooling layer was not used in our CNN model because all values of the output of the convolution layer were informative in our prediction. In this regard, a recent genetics analysis reported that amino acids distal to contact interfaces can exert significant effects on the interactions between MHC-peptide complexes and TCRs[6], implying that to detect precise binding patterns, all values in the output of the convolution layer should be taken into account in the pooling process.

To the second convolution layer we attached a fully connected layer in which all neurons receive inputs from all outputs of the previous layer for integration of information. This fully connected layer performed ReLU($WX$), where $X$ is the input and $W$ is the weight matrix for the fully connected layer. The last layer, the sigmoid output layer, performed classification between binding and non-binding with the prediction scaling in the range of 0 ~ 1 based on the sigmoid function. In other words, the sigmoid output layer performed Sigmoid($WX$), where $X$ is the input and $W$ is the weight matrix for the sigmoid output layer.

We trained our model in the direction toward the minimisation of the objective function, which was defined as the sum of negative log likelihood (NLL) and regularisation terms intended for overfitting control. Specifically,

$$\text{Objective} = \text{NLL} + \lambda_1 ||W||_2^2 + \lambda_2 ||H^{-1}||_1, \quad (2)$$

$$\text{where NLL} = -\sum_s \sum_t \log(Y_t^s f_t(X^s) + (1 - Y_t^s)(1 - f_t(X^s))),$$

and $s$ indicates the index of training samples and $t$ indicates the index of interaction features. $Y_t^s$ is a 0 or 1 label for sample $s$ and interaction feature $t$. $f_t(X^s)$ represents the predicted probability output of the model for interaction feature $t$ given input $X^s$. We used a combination of multiple regularisation techniques typically used for training deep neural networks. L2 regularisation term $||W||_2^2$ was defined to be the sum of the squares of all weight matrix entries. $||H^{-1}||_1$ was defined to be the L1 norm of all output values of the last layer (fully connected layer) preceding the output layer. Additionally, the optimisation was subjected to regularisation constraints that for any layer $m$ and neuron $n$, $||W_m^n||_2 \leq \lambda_3$ or that the L2 norm of weights for any neuron must not be larger than a specified value. Hyperparameters we used in the model included the learning rate [0.001, 0.01, 0.1], number of kernels for the first and second layer [10, 30, 50], L1 and L2 regularisation parameter [0.0001, 0.001, 0.01], and momentum [0,1, 0.5, 0.9]. Especially, various filter sizes (1~5 bp for peptides and 1/2, 2/3, and full length of HLAs) were used to extract interaction features in the convolution layers.

Derivatives of the objective function with respect to the model parameters were computed by the standard backpropagation algorithm. We optimised the objective function by using stochastic gradient descent with momentum. We did not use dropout training because it could cause a decrease in training performance. Our model was implemented using the Theano library (https://github.com/Theano/Theano/) on Tesla K40x GPU.

We ran HLAminer[37] for the samples of SMC cohort. We used HLA allele information provided by the authors for the published cohort samples. Amino acid sequences flanking nonsynonymous mutations were retrieved from RefSeq database[38] by using the idfetch programme. The HLA sequences and mutant peptide sequences were subjected to our CNN prediction model.

**SMC cohort for checkpoint blockade in lung cancer**. A total of 122 advanced non-small cell lung carcinoma (NSCLC) patients who were treated with anti-PD-1/PD-L1 from 2014 to 2017 at Samsung Medical Center were enroled for this study. The clinical response was evaluated by the Response Evaluation Criteria in Solid Tumours (RECIST) version 1.1 with a minimum 6-month follow-up. The response to immunotherapy was classified into durable clinical benefit (DCB, responder) or non-durable benefit (NDB, non-responder)[24]. Partial response (PR) or stable disease (SD) or that lasted more than 6 months was considered as DCB/responder. Progressive disease (PD) or SD that lasted less than 6 months was considered as NDB/non-responder. Progression-free survival (PFS) was calculated from the start date of therapy to the date of progression or death, whichever is earlier. Patients were censored at the date of the last follow-up for PFS if they were not progressed and alive. This study was approved by the institutional review board of Samsung Medical Center (2018-03-130 and 2013-10-112). Informed written consent was obtained from all patients enroled in the study.

Tumour samples were obtained before anti-PD1/PD-L1 treatment, and then were embedded in paraffin after formalin fixation or kept fresh. DNA was prepared using AllPrep DNA/RNA Mini Kit (Qiagen, 80204), AllPrep DNA/RNA Micro Kit (Qiagen, 80284), or QIAamp DNA FFPE Tissue Kit (Qiagen, 56404) for library preparation for whole exome sequencing. Library preparation was performed by using SureSelectXT Human All Exon V5 (Agilent, 5190–6209) according to the instructions[39]. Briefly, 200–300 ng of tumour and normal genomic DNA was sheared, and 150–200 bp of the sheared DNA fragments were further processed for end-repairing, phosphorylation, and ligation to adaptors. Ligated DNA was hybridised using whole-exome baits from SureSelectXT Human All Exon V5. The libraries were quantified by Qubit and 2200 Tapestation, and sequenced on an Illumina HiSeq 2500 platform with 2 × 100 bp paired ends. Target coverage for normal samples was 50 x and tumour sample was 100×. The sequencing reads were aligned to the hg19 reference genome by using Burrows-Wheeler Aligner (BWA) (http://bio-bwa.sourceforge.net)[40] version 0.7.5a. Genome Analysis Toolkit (GATK) (https://software.broadinstitute.org/gatk)[41] version 3.5 was applied for base quality score recalibration, indel realignment, and duplicate removal. The BAM files produced after these processes were subjected to MuTect (https://software.broadinstitute.org/cancer/cga/mutect)[42] version 1.1.4 for the calling of single nucleotide variants (SNVs), and small insertions and deletions (indels). The dbSNP and COSMIC databases were used as references. Data processing and analysis were done with default parameters. The calling results for SNVs and indels are provided in Supplementary Data 4. To estimate tumour purity, Clonal Heterogeneity Analysis Tool (CHAT) was used[43]. CNVkit[44], a copy number detection tool specific for whole-exome and short-read sequencing platforms, was used to detect copy number alterations between matched normal and tumour samples. The log2 coverage depth was defined as copy number variation (CNV) value. This indicates the ratio of the mean coverage depths, which is excluding extreme outliers and is observed at the corresponding bin in each sample. The CNV results for our cohort samples are provided in Supplementary Data 5. The results of neoantigen calling for the SNVs and indels from our SMC cohort samples are provided in Supplementary Data 6.

**Other cohorts for checkpoint blockade in lung cancer and melanoma**. Somatic mutation calls for samples in three independent melanoma cohorts with anti-CTLA-4, namely, Van allen et al.'s dataset[21], Snyder et al.'s dataset[22], and Roh et al.'s dataset[10], one melanoma cohort with anti-PD-1, namely, Riaz et al.'s dataset[23], two NSCLC cohorts with anti-PD-1, namely Rizvi et al.'s dataset[24], Hellmann et al.'s dataset[25] were collected. For each cohort, we used the other cohorts of the same tumour type as training data. For example, we trained random forests with Hellman and Rizvi cohorts and tested performance on SMC cohort. The AUCs were 0.81 ~ 0.97 for the training data and 0.76 ~ 0.95 for the test data. We used HLA allele information provided for each sample. Amino acid sequences flanking nonsynonymous mutations were retrieved from RefSeq database[38] by using the idfetch programme. The HLA sequences and mutant peptide sequences were subjected to our CNN prediction model. Resistant samples were defined as having >70 predicted neoantigens[9] while no clinical benefit was reported from each respective study. All remaining samples were defined as non-resistant samples and trained along with the resistant samples.

**TCGA data processing**. We downloaded TCGA RNA-Seq data from dbGaP (https://www.ncbi.nlm.nih.gov/gap). HLA typing was performed by applying Seq2HLA tool[45] to the RNA-Seq data. Somatic mutation calls were obtained from the UCSC Xena Browser (http://xena.ucsc.edu). Amino acid sequences flanking nonsynonymous mutations were retrieved from RefSeq database[38] by using the idfetch programme. The HLA sequences and mutant peptide sequences were subjected to our CNN prediction model. We also profiled the TCR repertoire and calculated the immune score for each TCGA sample. The TCR repertoire was obtained by applying the TRUST tool[26] to the RNA-Seq data. The immune score was calculated as the geometric mean of the expression level of cytolytic markers (GZMA, GZMB, PRF1, and GNLY), HLA molecules (HLA-A, HLA-B, HLA-C, HLA-E, HLA-F, HLA-G, HLA-H, HLA-DMA, HLA-DMB, HLA-DOA, HLA-DOB, HLA-DPA1, HLA-DPB1, HLA-DQA1, HLA-DQA2, HLA-DQB1, HLA-DRA, and HLA-DRB1), IFN-γ pathway genes (IFNG, IFNGR1, IFNGR2, IRF1, STAT1, and PSMB9), chemokines (CCR5, CCL3, CCL4, CCL5, CXCL9, CXCL10, and CXCL11), and adhesion molecules (ICAM1, ICAM2, ICAM3, ICAM4, ICAM5, and VCAM1)[10]. We used log2-transformed the normalised RNA-seq read counts. We also selected the molecular markers that were best correlated with neoantigen load in each tumour type: CCL5 and IFNG for BLCA; CD247, ICAM2, and IFNGR2 for ESCA; GZMB, GZMH, and PRF1 for HNSC; CCL4, CCR5, CXCL11, CXCL9, and GZMH for LUSC. For samples with potential immune evasion, we detected tumours with high (higher then 70th percentile) neoantigen load and low (lower than 30th percentile) immune signature defined by the average of the selected immune markers.

**Patient survival analysis**. We utilised the number of neoantigens or mutations as a predictor to perform patient survival analysis. For the clinical trial data, the cases in which the number of neoantigens or mutations was equal to or greater than 70 were classified to the high neoantigen or mutation group as suggested in a previous study[9]. The melanoma samples of the two clinical trials anti-CTLA-4[21,22] and the lung cancer samples of three clinical trials anti-PD-1 or anti-CTLA-4[24,25] were subjected to survival analysis. The patients who died for reasons other than tumour were excluded from the analysis. The skin cutaneous melanoma samples from TCGA were divided into two groups based on whether the number of neoantigens or mutations was higher or lower than the median level. P values from the Wald test were used to determine the significance of differences between two groups.

**Mutational analysis of therapeutic resistance**. For each of all cohorts used in this study (Table 1), resistant samples were defined as having > 70 predicted neoantigens[9] while no clinical benefit was reported from each respective study. All remaining samples were used as control. Functional mutations were called in each sample. Mutation functionality was defined based on SIFT (http://sift.bii.a-star.edu.sg)[46] and PROVEAN (http://provean.jcvi.org)[47]. The mutations that were called simultaneously as damaging by SIFT and as deleterious by PROVEAN were defined as functional mutations. We then constructed a matrix for mutation status of genes for which the mutation frequency was >5% in the given population of samples. A random forest predictor consisting of 1000 decision trees was trained for tumours with therapeutic resistance. We implemented random forests by running the R package randomForest[48] with ten-repeat 5-fold cross validation. We used the status of synonymous mutations on the same set of genes as a negative control training model. For each training, the number of features (mutated genes) was set to identical between the original and negative control model. The threshold of the mutation frequency was adjusted so that the same number of features were used.

The 'variable importance' of each feature (mutated gene) was evaluated on the basis of the mean decrease in accuracy as implemented in the randomForest R package. Specifically, the importance of the kth feature was measured as the degree of decrease in prediction accuracy upon random permutation of all values in the kth feature of the training dataset. We then performed a pathway analysis for the genes whose variable importance was greater than 3 and their interaction partners in the protein interactome. We used an integrative interactome map encompassing an integrated physical interaction network, referred to as Interactome, created by merging yeast two-hybrid-based proteome-scale interacting pairs[27], integrated literature-based protein-protein interactions[27], binary interactions identified from Stitch-seq interactome mapping[29], and interactions from the high-quality protein interactome from the HINT database[28]. Enrichr[49] was used to analyse the extended gene on the basis of pathways and functional terms retrieved from Reactome[50], Panther[51], and Gene Ontology[52]. Panther[51] was performed with its original pathway analysis algorithm. Additional gene ontology terms that were not found in Enrichr[49] were retrieved by using DAVID[53] GOTERM_BP_DIRECT.

**Regression analysis for resistance or immune evasion**. We used SMC cohort and Roh cohort for the quantitative analysis of the explanatory power of our exomic prediction in comparison with previously known resistance parameters. The resistance parameters reported in the literature were obtained from the data of SMC cohort. Roh cohort was used because copy number alteration (gain or loss) and tumour heterogeneity were available. For the TCGA samples set apart as testing data, we employed the SCNA level calculated in a previous work[11]. We

performed regression analysis for resistance or immune evasion on these resistance parameters together with our mutation based-prediction scores. The raw values of the resistance parameters were scaled from 0 to 1 as

$$\text{scaled}(e_i) = \frac{e_i - E_{min}}{E_{max} - E_{min}} \quad (3)$$

where $e_i$ indicates the $i^{th}$ value of variable $E$. $E_{min}$ and $E_{max}$ indicate the minimum and maximum value for variable $E$, respectively. Our mutation based-prediction score was given as the vote ratio ranging from 0 to 1 that was provided by the random forest predictor. Therefore, this score was used without scaling. We then performed multiple linear regression by the simple, lasso, elastic net, and ridge regression method with 5-fold cross validation implemented in the glmnet R package[54]. The ROCR[55] R package was used to calculate the AUC to assess the accuracy of each regression model. The variable importance of each feature in the regression model was evaluated on the basis of generalised cross-validation (GCV) implemented by the glmnet R package.

**Reporting summary**. Further information on research design is available in the Nature Research Reporting Summary linked to this article.

## Data availability

All data including clinical information and mutation/neoantigen calls for our checkpoint blockade cohort are provided as Supplementary Data. The source data underlying Fig. 1–5 are provided as a Source Data file.

## Code availability

Codes for implementing the CNN model to predict peptide-MHC class I binding were made available at the authors' webpage (http://omics.kaist.ac.kr/resources).

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

## Acknowledgements

This work was supported by the Post-Genome Technology Development Program [10067758] funded by the Ministry of Trade, Industry and Energy, by the Bio & Medical Technology Development Program (NRF-2019M3A9B6064688) funded by the Ministry of Science and ICT, and by the Basic Science Research Program [2018R1D1A1B07047485] funded by the Ministry of Education .

## Author Contributions

K.K. performed all data analyses and wrote the manuscript. H.S.K. generated cohort data and helped with manuscript writing. J.Y.K. and H.J. participated in data analyses. J.-M.S., J.S.A., M.-J.A., and K.P. helped with cohort data generation and analyses. S.-H.L. and J.K. C. conceived and supervised the study.

## Competing interests

The authors declare no competing interests.
