## [Peer Review File · Nature Communications]

Reviewers' comments:

Reviewer #1 (Remarks to the Author):

General comments

The manuscript by Kim et al. describes the development of a predictor for MHC-binding neoantigens based on convolutional neural network and its application to several datasets including two cohorts treated with anti-CTLA4 antibodies and several TCGA datasets. The authors suggest that the neoantigen load is a predictor of response to immune checkpoint blockade and that functional immune mutations explain poor therapeutic response.

The manuscript does not provide novel insights and has major methodological flaws. It has been previously shown that mutational load is associated with response to immune checkpoint blockade albeit the predictive power is not very strong. Since mutational load is highly correlated with neoantigen load, this is not a novel finding. There are also major methodological flaws with respect to the testing and selection of the tools (see also specific comments). Most importantly, the authors compared the performance of their algorithm with an old version of NetMHCpan (published in 2007). There are several improvements of NetMHCpan since then, the latest being netMHCpan 4.0 (Jurtz et al., J Immunol 2017) that was trained on binding affinity and eluted ligand data. It was shown also in another study (Bassani-Sternberg et al., PLoS Comp Biol 2017) that predictors trained on MS data show very high predictive power for the identification of neoantigens.

Specific comments

1. After the installation the tool didn't work so the results could not be replicated. The example code proposed in the README file

```
python run_cnn.py data/example.npz result/example_result.txt
```

delivered the following error:

Traceback (most recent call last):

```
File "cnn.py", line 12, in <module>
```

```
    predFile = sys.argv[3]
```

```
IndexError: list index out of range
```

2. The analysis of the TCR repertoires should be carried out with an accurate tool which extracts nearly zero CDR3-like false positives (MiXCR, Bolotin et al., Nat Biotech 2017).

3. Discussion, first paragraph: the authors discuss tumor heterogeneity but do not provide any data. Analysis of the tumor heterogeneity should be carried out with an appropriate tool (see paper by McGranahan et al, Science 2016).

Reviewer #2 (Remarks to the Author):

In this manuscript, Kim and colleagues make two novel contributions: (1) they develop a new MHC Class I peptide binding algorithm based on convolutional neural networks and (2) they suggest that mutations in immune-related genes in high mutation load tumors may mediate resistance to checkpoint blockade therapy. Both of these findings are interesting and merit publication – however some technical details regarding the methods require further clarification before this manuscript would be acceptable for publication as discussed below.

1.) The description of the construction of the convolutional neural-network needs more clarity. Its

not clear to me how the 9-mer (or 10-mer) and the HLA-sequence are both being adequately coded in the input layer. Figure 1a is not sufficient for this purpose, and this may require an additional supplemental figure to clarify. The sentence “We employed amino acid interaction preferences computed based on contacts between Ca atoms or between any atoms in native protein structures” in the methods also is unclear. The authors can refer to another preprint on a deep CNN for peptide prediction that is a bit more clearly described at:

<https://www.biorxiv.org/content/biorxiv/early/2017/12/24/239236.full.pdf>

2.) Description of the test sets used to evaluate the performance of the CNN should be clearer (i.e. how many peptide bindings were tested for each data-set). Figure 1B would be better displayed as showing AUCs for all 4 methods for selected HLA-A and HLA-B alleles clearly (instead of multiple panels for each method individually) --- this type of raw data could be supplied as table or a supplementary figure. The 70.5% superior performance really should be clarified by comparing the CNN to each method individually (how often is the authors method better than netMHCpan, then SMN, etc.)

3.) “The P value of our association was 5×10^{-3} for Van allen et al.’s dataset, for which a marginal association ($P = 0.027$) was observed when NetMHCpan was used for neoantigen prediction” --- Please specify which test was used to determine p-values (assume Wilcoxon rank-sum test)

4.) “Higher neoantigen burden was significantly associated with longer disease free time consistently in the two cohorts (Fig. 2B).” – how was higher neo-antigen burden determined? (Was this cut at the median, or > 70 neo-antigens – please include this detail in figure legend?)

5.) “This correlation was higher than when mutational load was used (Supplementary Fig. 5)” – what cutpoint was used here for high vs. low mutation load.

6.) The 269 genes that were ultimately selected ---- are these only related to immune functions, or were these selected from all 20,000 genes to start?

7.) Would you not expect that NSCLC (Figure 3B) also would develop resistance to checkpoint blockade by developing mutations in immune-related genes like melanomas? i.e. why is the the NSCLC data really a negative control?

Minor Points

- Introduction would benefit from clearly defining neo-antigen as a couple of slightly different definitions exists (i.e. explicitly stating these arise from mutations in the cancer genome that produce novel peptides)
- The datasets used for training the authors CNN and the tests sets should be provided with the code to facilitate reproducibility.
- Multiple metrics can be used to quantify TCR Diversity (Shanon entropy, evenness, etc.. please clarify which metric is being used in 2C)
- Figure references are missing in main text for Figure 3A
- 2D only shows a weak trend for survival
- The interpretation of “variable importance” in figure 4B-D is unclear – should be clarified in caption legend
- Lastly, although mostly well written, the manuscript would benefit from additional English-language editing to facilitate clarity.

Reviewer #3 (Remarks to the Author):

The paper describes two disparate methods that are related via immunotherapy applications. The first method uses convolutional neural networks to predict MCH binding peptides. The second uses random forests trained on point mutations to predict resistance to checkpoint blockade. The claim is made that combining these two approaches "accurately predicts therapeutic response," but the current presentation of the data does not support this conclusion.

"To build an amino acid interaction map for our CNN model, we inferred the binding preference of each pair of amino acids using interaction energy estimated from the frequency of neighboring amino acids in native protein structures"

A few sentences should be included explaining how these interaction energies are computed - this is too important a detail to relegate to a reference. Structures do not exist for all considered peptide/MHC pairs so how are interactions determined?

For all classification datasets the number of true and false examples should be reported.

The description of the CNN model lacks detail. The number, dimension, and stride of the kernels should be reported. The text states the "convolution layers of our model performed one-dimensional convolution" but Fig. 1 shows a 2D convolution. Were multiple models trained and an ensemble used to predict, or only a single model? What is the variance across models trained using different random seeds?

Is the sequence itself part of the input to the CNN, or only the interaction energies?

How well do the trained models fit the training set vs the test set? This would be useful to have in the supplement (gives an idea of overfitting).

Fig 1B. "70.5% of the test datasets (100/132)" I believe this is incorrectly worded and there are only 33 test datasets, which were evaluated using 4 different comparison models. This needs to be reworded (e.g. "outperforms alternative approaches in X to Y% of the datasets).

The application of 5-fold cross-validation in training the RF model is not adequately explained. Was this used to set the hyperparameters for training on the full dataset? Was it somehow used to create the model that was evaluated on the test set?

Were the 269 genes all the genes "that harbour deleterious or damaging mutations" as described in the text, or was this number determined through cross-validation? The number of features (269) is greater than the number of training examples (174). Unless there are a lot of correlated features, this would lead to overfitting if no regularization is performed.

The 100 tests are not explained at all in the results section. What is the size of the 100 subsets chosen? Are they sampled with or without replacement? This sort of analysis gives an estimate of variance, not "predictive performance" as stated in the text.

"accuracy > 3 for the clinical melanoma data and accuracy > 1 for the TCGA". Note sure what is being communicated here. How is accuracy being measured? Are those suppose to be percentages?

Fig. 4. There are quite a few problems here.

"Of the 46 non-responders, our CNN method predicted a low load of neoantigens for 15 samples (grey bars in the upper graph of Fig. 4A) and a high load of neoantigens for 31 samples (blue and orange bars in the upper graph of Fig. 4A)."

"(A) Neoantigen load and resistance parameters for 47 nonresponders to checkpoint blockade in Roh et al.'s cohort"

46 != 47 and there are actually 52 datapoints in the figure. Roh's dataset has 56 patients. There is a "Responder" and a "Non-responder" label on the figure, but it isn't at all clear what they are trying to communicate and the text of the paper makes no reference to the responders.

This is unfortunate, since in order to justify the claim that a method has predictive power, it must be able to distinguish between the two classes of interest. Perhaps the first six bars represent the responders? But each of these examples has a comparable example among the non-responders - there is no clear discrimination visible in this figure.

"Two bar plots below depict neoantigen load estimated by the CNN model and NeMHCpan." There is only one bar plot (CNN) in 4A

"The grey bars mark the cases in which no therapeutic response is supported by low neoantigen load." Why are some grey bars above the drawn threshold line then?

What is a mutation signature? It isn't defined. If it is the score produced by the RF, that isn't a "signature," that's a single number. What threshold is used to draw a green box in Fig 4A?

"from simple logistic regression, the mutation signatures only were capable of predicting resistance at high accuracy when our CNN model was used for neoantigen identification (Fig 4B)"

"AUC (upper) and variable importance (lower) for the regression of resistance on the mutation signatures when resistance was defined by neoantigen load estimated by CNN or NetMHCpan."

This is confusing and misleading to the point I'm not sure what 4B,C,D are showing. Based on the results section, this should be showing that "profiling functional mutations together with neoantigens accurately predicts therapeutic response," but based on the caption text, it is showing that the functional mutations (and other indicators) can be fit in a linear model to the _predictions_ of neoantigen level, which is not the same.

I believe the caption is correct (since the neoantigen score is not shown with a weight in the heatmaps) which means, unfortunately, there is no combining of the two approaches presented.

Equating "resistance" with the neoantigen level seems wrong, especially considering Fig 4A. Some explanation of why this exercise is informative/useful would be appreciated.

The selection of y-axis scales in 4B,C,D is inconsistent and misleading.

An AUC doesn't make much sense for a regression model - it is a measure of classification accuracy. Why would this be used?

What is needed is evidence the two approaches can be combined to predict therapeutic response.

Reviewer #1 (Remarks to the Author):

General comments

The manuscript by Kim et al. describes the development of a predictor for MHC-binding neoantigens based on convolutional neural network and its application to several datasets including two cohorts treated with anti-CTLA4 antibodies and several TCGA datasets. The authors suggest that the neoantigen load is a predictor of response to immune checkpoint blockade and that functional immune mutations explain poor therapeutic response.

The manuscript does not provide novel insights and has major methodological flaws. It has been previously shown that mutational load is associated with response to immune checkpoint blockade albeit the predictive power is not very strong. Since mutational load is highly correlated with neoantigen load, this is not a novel finding. There are also major methodological flaws with respect to the testing and selection of the tools (see also specific comments). Most importantly, the authors compared the performance of their algorithm with an old version of NetMHCpan (published in 2007). There are several improvements of NetMHCpan since then, the latest being netMHCpan 4.0 (Jurtz et al., J Immunol 2017) that was trained on binding affinity and eluted ligand data. It was shown also in another study (Bassani-Sternberg et al., PLoS Comp Biol 2017) that predictors trained on MS data show very high predictive power for the identification of neoantigens.

We thank the referee for reviewing our manuscript and providing constructive and valuable comments. First of all, please note that the correlation of neoantigen load with the clinical response is not our main finding. This association was examined simply to support the validity of our MHC-binding prediction method. Our major finding for which we claim novelty is that increased mutation burden can be responsible for functional mutations that facilitate immune evasion, and that this aspect can be more clearly observed with our accurate MHC-binding prediction.

We agree with the reviewer that there were weak points in our methodologies, and so we made considerable efforts to correct those flaws. First, we collected additional clinical data including our own cohort (revised Table 1), referred to as SMC (Samsung Medical Center) cohort, to validate the consistency of our method.

Cohort name	Tumor type	Cohort size	Target checkpoint	Reference
SMC	Lung cancer	122	PD-1/PD-L1	This work
Hellmann	Lung cancer	75	PD-1 & CTLA-4	Ref 25

Rizvi	Lung cancer	34	PD-1	Ref 24
Van Allen	Melanoma	110	CTLA-4	Ref 21
Snyder	Melanoma	64	CTLA-4	Ref 22
Roh	Melanoma	56	PD-1 & CTLA-4	Ref 10
Riaz	Melanoma	68	PD-1	Ref 12

As suggested by the reviewer, we used NetMHCpan 4.0 (Jurtz et al., J Immunol 2017) for comparison with our CNN model. We also used the latest versions of other algorithms (e.g., NetMHCcon) for comparison. The comparison results, shown below, are provided in revised Figure 1B.

During the comparison tasks, we detected undesirable variabilities in the performance levels when the size of training data was not sufficiently high. Therefore, only the HLA alleles with the size of training data > 3,000 were used for performance comparison. In the same context, we expect the training-set size of newly generated MS data (Bassani-Sternberg et al., PLoS Comp Biol 2017) will increase soon to the level at which deep learning algorithms can have competitive edge over other machine learning algorithms.

As pointed out by the referee, mutation burden or neoantigen load cannot fully predict the therapeutic response of immune checkpoint blockade. However, our neoantigen prediction showed a better correlation with the clinical response and patient survival in multiple melanoma and lung cancer cohorts (revised Figure 2A, 2B and revised Supplementary Figure 5; attached below). The survival analysis was performed for the five cohorts in which neoantigen load showed a significant correlation with the clinical benefit.

<Predictive power of neoantigen load estimated by CNN and NetMHCpan 4.0>

<Patient survival stratified by CNN neoantigen load>

<Patient survival stratified by NetMHCpan neoantigen load>

We then used our CNN method to select patient samples with high neoantigen load and examined what makes some of the samples resistant to checkpoint blockade. For this purpose, we developed a prediction algorithm based on functional mutations that alter protein functions. This novel method was able to predict the therapeutic resistance at substantially high accuracy consistently across different training and testing data (revised Figure 3A) as attached below.

We therefore propose that this combination of antigenic mutations (neoantigens) and functional mutations can predict patient response better. This is the main finding for which we claim novelty.

Specific comments

1. After the installation the tool didn't work so the results could not be replicated. The example code proposed in the README file python run_cnn.py data/example.npz result/example_result.txt delivered the following error:

Traceback (most recent call last):
File "cnn.py", line 12, in <module>
predFile = sys.argv[3]
IndexError: list index out of range

We apologize for the inconvenience by the error. Now all the errors have been corrected. We have uploaded the updated codes to our webpage under the link named "Predicting clinical benefit of immunotherapy by antigenic or functional mutation affecting tumor immunogenicity" (<https://omics.kaist.ac.kr/resources>).

2. The analysis of the TCR repertoires should be carried out with an accurate tool which extracts nearly zero CDR3-like false positives (MiXCR, Bolotin et al., Nat Biotech 2017).

We thank the reviewer for this constructive comment. We analyzed the TCR repertoires by the recommended tool (MiXCR, Bolotin et al., Nat Biotech 2017). As shown below (revised Figure 2C), higher neoantigen load correlated with higher TCR diversity in agreement with the previous results.

3. Discussion, first paragraph: the authors discuss tumor heterogeneity but do not provide any data. Analysis of the tumor heterogeneity should be carried out with an appropriate tool (see paper by McGranahan et al, Science 2016).

Tumour heterogeneity (analyzed by PyClone released by McGranahan et al, Science 2016 and by SciClone released by Miller et al, PLoS Comput Biol 2014) was used as one of the resistance factors in our regression analysis. These tools have been cited as shown below:

(Page 9, Line 31) We next wanted to evaluate the clinical utility of combining these two approaches on the basis of the melanoma samples in Roh cohort and lung cancer samples in SMC cohort that came with information on therapeutic response to checkpoint inhibitor and known resistance parameters such as copy number alteration (gain or loss) and tumour heterogeneity⁹⁻¹¹.

These tools are denoted as H1 and H2 in revised Figure 4 (see the red boxes) as attached below.

EP = Exomic prediction, J = JAK mutation, E = EGFR mutation, I = Immune-related pathway mutation, A = Antigen-presenting pathway mutation, G = Copy number gain, L = Copy number loss, H1 = Heterogeneity by Expands, H2 = Heterogeneity by SciClone

EP = Exomic prediction, S = SERPINB mutation, J = JAK mutation, E = EGFR mutation, B = B2M mutation, I = Immune-related pathway mutation, A = Antigen-presenting pathway mutation, G = Copy number gain, L = Copy number loss, H1 = Heterogeneity by Expands, H2 = Heterogeneity by SciClone

Reviewer #2 (Remarks to the Author):

In this manuscript, Kim and colleagues make two novel contributions: (1) they develop a new MHC Class I peptide binding algorithm based on convolutional neural networks and (2) they suggest that mutations in immune-related genes in high mutation load tumors may mediate resistance to checkpoint blockade therapy. Both of these findings are interesting and merit publication – however some technical details regarding the methods require further clarification before this manuscript would be acceptable for publication as discussed below.

1. The description of the construction of the convolutional neural-network needs more clarity. Its not clear to me how the 9-mer (or 10-mer) and the HLA-sequence are both being adequately coded in the input layer. Figure 1a is not sufficient for this purpose, and this may require an additional supplemental figure to clarify. The sentence “We employed amino acid interaction preferences computed based on contacts between C α atoms or between any atoms in native protein structures” in the methods also is unclear. The authors can refer to another preprint on a deep CNN for peptide prediction that is a bit more clearly described at: <https://www.biorxiv.org/content/biorxiv/early/2017/12/24/239236.full.pdf>

We thank the reviewer for acknowledging the novelty and value of our contributions. We also appreciate the valuable comments on our description. The input layer was constructed based on the matrix of amino acid interactions between a 9-mer (or 10-mer) peptide and a 365-mer HLA. This matrix is filled by the values of interaction preferences for all pairs of amino acids between the given peptide and HLA. The structure of our input layer is completely different from the CNN structure mentioned by the reviewer (JianJun Hu et al. <https://doi.org/10.1101/239236>), in that the published CNN algorithm did not use information of amino acid interactions at all but only the sequences themselves as done in NetMHCpan.

For the assessment of interaction preferences among amino acids between peptides and HLA proteins, we capitalized on a published dataset for the contact frequencies among 20x20 amino acid pairs in single protein structures (Vishveshwara, S. et al. Protein Sci 2010). All peptide-HLA interaction matrices could be filled based on the map of 20x20 amino acid interaction preferences. We provide this information in revised Supplementary Figure 1 as seen below.

A Amino acid interaction preference map

B HLA sequence (365-mer)

C HLA sequence (365-mer)

	Y	Y	A	K	R	R	W	A	.	.	Q	S
D	-0.02	-0.02	-0.02	-0.06	-0.06	-0.06	0.00	-0.02				Kernel A ₁
E	-0.02	-0.02	0.00	-0.07	-0.07	-0.07	-0.01	0.00				Kernel A ₁
F												
.												
.												
.												
L												Kernel A _n
R												Kernel A _n

2. Description of the test sets used to evaluate the performance of the CNN should be clearer (i.e. how many peptide bindings were tested for each data-set). Figure 1B would be better displayed as showing AUCs for all 4 methods for selected HLA-A and HLA-B alleles clearly (instead of multiple panels for each method individually) --- this type of raw data could be supplied as table or a supplementary figure. The 70.5% superior performance really should be clarified by comparing the CNN to each method individually (how often is the authors method better than netMHCpan, then SMN, etc.)

The number of peptides for each test data set was described in revised Supplementary Figure 3 (attached below), which also included the number of positive (binding) peptides and the ratio of positive peptides. Please note that we detected undesirable variabilities in the performance levels when the size of training data was not sufficiently high. Therefore, only the HLA alleles with the

size of training data > 3,000 were used for performance comparison. Shown below are the corresponding test datasets.

We rearranged the results of performance comparison as shown in revised Figure 1B (attached below).

Furthermore, each comparison between our CNN and other tools is described in our revised manuscript as follows:

(Page 5, Line 18) In terms of AUC, our method was superior to SMPMBEC, ANN, NetMHCcon, and NetMHCpan for 100%, 100%, 90%, and 70% of the test cases. With regards to the F1 score, our CNN was superior to all the methods for 80% of the test cases.

3. “The P value of our association was 5×10^{-3} for Van allen et al.’s dataset, for which a marginal association ($P = 0.027$) was observed when NetMHCpan was used for neoantigen prediction” --- Please specify which test was used to determine p-values (assume Wilcoxon rank-sum test).

We used the Wilcoxon rank-sum test as assumed by the reviewer. The specified method is now shown in the legend of revised Figure 2A.

4. “Higher neoantigen burden was significantly associated with longer disease free time consistently in the two cohorts (Fig. 2B).” – how was higher neo-antigen burden determined? (Was this cut at the median, or > 70 neo-antigens – please include this detail in figure legend?)

We extended the analysis of the association between disease-free time and neoantigen burden to include five clinical cohorts as shown in revised Figure 2B (attached below). We set the cutoff of the neoantigen count to 70 consistently across all the cohorts. We added this description in the figure legend of revised Figure 2B as shown below (see the red box).

5. “This correlation was higher than when mutational load was used (Supplementary Fig. 5)” – what cutpoint was used here for high vs. low mutation load.

The statement was about the results from two melanoma cohorts (Van Allen and Snyder). During revision, we substantially extended the data to cover seven cohorts in melanoma and lung cancer (revised Table 1 attached below). We performed survival analysis for five cohorts in which neoantigen load showed a significant correlation with the clinical response.

Cohort name	Tumor type	Cohort size	Target checkpoint	Reference
SMC	Lung cancer	122	PD-1/PD-L1	This work
Hellmann	Lung cancer	75	PD-1 & CTLA-4	Ref 25
Rizvi	Lung cancer	34	PD-1	Ref 24
Van Allen	Melanoma	110	CTLA-4	Ref 21
Snyder	Melanoma	64	CTLA-4	Ref 22
Roh	Melanoma	56	PD-1 & CTLA-4	Ref 10
Riaz	Melanoma	68	PD-1	Ref 12

We discovered extremely high variability in mutation burden across the cohorts, thereby making it hard to use as a predictive variable. In contrast, neoantigen load either calculated by NetMHCpan or CNN showed reasonable intercohort variations, thereby enabling us to set a threshold (high neoantigen load defined as the neoantigen count > 70). The comparisons between CNN and NetMHCpan are provided in revised Figure 2B (CNN) and Supplementary Figure 5 (NetMHCpan) as attached below.

CNN

NetMHCpan

6. The 269 genes that were ultimately selected ---- are these only related to immune functions, or were these selected from all 20,000 genes to start?

The 269 genes were selected from all 20,000 initial genes. In fact, we attempted to use mutations on immune-related genes only, but the mutation frequency of

these genes was not relevant to the clinical response as described in revised Supplementary Figure 6. We thus selected genes that played a critical role in predicting the clinical response during machine learning by random forests.

7. Would you not expect that NSCLC (Figure 3B) also would develop resistance to checkpoint blockade by developing mutations in immune-related genes like melanomas? i.e. why is the the NSCLC data really a negative control?

The reviewer was correct in that NSCLC also would develop resistance to checkpoint blockade. During revision, we substantially increased the amount of data to test lung cancer cohorts in the same manner as the melanoma cohorts. We also generated our own lung cancer cohort (n = 122). For each cohort, we used the other cohorts of the same tumor type as training data. For example, we trained random forests with Hellman and Rizvi cohorts and tested performance on SMC cohort. As a result, our prediction model was capable of predicting therapeutic resistance in both lung cancer and melanoma. The synonymous mutation models were used as negative control. We updated these results in revised Figure 3A as attached below.

Minor Points

- Introduction would benefit from clearly defining neo-antigen as a couple of slightly different definitions exists (i.e. explicitly stating these arise from mutations in the cancer genome that produce novel peptides)

We modified the Introduction section as follows:

(Page 3, Line 14) Tumour neoantigens are generated by somatic mutations producing novel peptides that can be recognized as foreign, thereby conferring immunogenicity to cancer cells. Neoantigen burden is therefore regarded as a fundamental determinant of response to immunotherapy including checkpoint blockade.

- The datasets used for training the authors CNN and the tests sets should be provided with the code to facilitate reproducibility.

The training data were embedded in the provided prediction model. To address the reviewer's comment, we have now separately attached the training and test datasets for our CNN model to the source codes linked under "Predicting clinical benefit of immunotherapy by antigenic or functional mutation affecting tumor immunogenicity" at our webpage (<https://omics.kaist.ac.kr/resources>).

- Multiple metrics can be used to quantify TCR Diversity (Shanon entropy, evenness, etc.. Please clarify which metric is being used in 2C)

We calculated TCR diversity by using the computational method called TRUST which directly infers immune repertoires from unselected bulk tumor RNA-seq data. In this analysis, different CDR3 sequences were counted as TCR diversity without using metrics such as Shannon entropy and evenness. We also analyzed TCR diversity by using another computational tool named MiXCR (Bolotin et al., Nat Biotech 2017), and the results were consistent with TRUST as seen below (revised Figure 2C).

- Figure references are missing in main text for Figure 3A.

Corrected.

- 2D only shows a weak trend for survival

Fig. 2D shows a weak trend for survival because this result was derived from TCGA samples but not immunotherapy cohort samples. With limited immune pressure, it is expected that the effects of neoantigen load will not be as evident as in the clinical setting of immunotherapy. Our focus is that despite this, neoantigen load estimated by our method has better explanatory power than mutation burden. We revised the description of Fig. 2D as follows:
 (Page 6, Line 29) For these samples, our estimate of neoantigen load than mutation burden showed a better correlation with patient survival, albeit with limited statistical significance (Fig. 2D).

- The interpretation of “variable importance” in figure 4B-D is unclear – should be clarified in caption legend.

We clarified “variable importance” in the caption legend of revised figure 4B-F as follows:

<Figure 4B, C>

<Figure 4E, F>

- Lastly, although mostly well written, the manuscript would benefit from additional English-language editing to facilitate clarity.

Our original manuscript has undergone editing by a native speaker. According to the reviewer's suggestion, we will send our manuscript out for professional English editing once our revision has gone through the second round of review and reached a decision for minor revision.

Reviewer #3 (Remarks to the Author):

The paper describes two disparate methods that are related via immunotherapy applications. The first method uses convolutional neural networks to predict MCH binding peptides. The second uses random forests trained on point mutations to predict resistance to checkpoint blockade. The claim is made that combining these two approaches "accurately predicts therapeutic response," but the current presentation of the data does not support this conclusion.

1. "To build an amino acid interaction map for our CNN model, we inferred the binding preference of each pair of amino acids using interaction energy estimated from the frequency of neighboring amino acids in native protein structures" A few sentences should be included explaining how these interaction energies are computed - this is too important a detail to relegate to a reference. Structures do not exist for all considered peptide/MHC pairs so how are interactions determined?

We thank the reviewer for providing constructive comments that we believe considerably improved our manuscript. We apologize for the lack of details about the interaction energies. We provide further descriptions on how the interaction energies are calculated and integrated into our model in the Method section and revised Supplementary Figure 1 as follows:

(Page 14, Line 26) We employed amino-acid interaction preferences computed based on contacts between C α atoms or between any atoms in native protein structures¹⁸. For the calculation of the interaction map, the dataset of 1,654 proteins from the PISCES server⁴⁰ was curated, and structural information from the PDB⁴¹ was obtained. The connectivity matrix based on the C α - C α distance was generated for each protein based on the distance cutoff of 6.5 Å between C α -C α atoms of amino acids with the exclusion of nearest neighbours along the sequence. Atom-atom contacts between two amino acid residues were also used such that residues i and j were considered to be in contact if any atom of the residue i is within a distance of 4.5 Å with any atom of the residue j. In this case, nearest neighbours ($i \pm 2$) along the sequence are not considered.

A Amino acid interaction preference map

B HLA sequence (365-mer)

C HLA sequence (365-mer)

	Y	Y	A	K	R	R	W	A	.	.	Q	S
D	-0.02	-0.02	-0.02	-0.06	-0.06	-0.06	0.00	-0.02				Kernel A ₁
E	-0.02	-0.02	0.00	-0.07	-0.07	-0.07	-0.01	0.00				Kernel A ₁
F												
.												
.												
.												
L												Kernel A _n
R												Kernel A _n

2. For all classification datasets, the number of true and false examples should be reported.

We used HLA alleles for which > 3,000 binding results were available. We provide the number of true (binding) and false (non-binding) cases in the training data in the Method section as follows:

(Page 14, Line 7) This database provided 57,173 data points consisting of binding affinity in terms of IC₅₀/EC₅₀ nM for 14,234 true (binding) and 42,879 false (non-binding) experiments.

The number and proportion of true positive, true negative, false positive, and false negative cases for each testing dataset is now provided in revised Supplementary Figure 3 (attached below).

3. The description of the CNN model lacks detail. The number, dimension, and stride of the kernels should be reported. The text states the "convolution layers of our model performed one-dimensional convolution" but Fig. 1 shows a 2D convolution. Were multiple models trained and an ensemble used to predict, or only a single model? What is the variance across models trained using different random seeds?

A variety of kernels were tested to achieve as high performance as possible. As pointed out by the reviewer, 2D convolution was used. We modified and added more details to the description of the CNN model in our revised manuscript as follows:

(Page 15, Line 17) A variety of kernels were tested to achieve as high performance as possible. The two convolution layers of our model performed 2D convolution operation after optimization of settings with 50 kernels for the first

layer, 10 kernels for the second layer, 1,000 batches, the stride size of 1, and the kernel size of 5x183 for each layer.

During our model training, multiple models from varying inputs (seeds) were tested. Comparison of the performance of the tested multiple models is provided in revised Supplementary Figure 2 as attached below.

4. Is the sequence itself part of the input to the CNN, or only the interaction energies?

Only the interaction energies without the sequences were taken as input to the CNN model. The sequence pairs between 365-mer HLAs and 9-mer peptides were used as templates to construct the 20x20 amino-acid map of interaction energies (Vishveshwara, S. et al. Protein Sci 2010). We now provide revised Supplementary Figure 1 to give more detailed information.

5. How well do the trained models fit the training set vs the test set? This would be useful to have in the supplement (gives an idea of overfitting).

All trained CNN and RF models fitted the training set better than the test set. However, the difference was marginal across all models. We provide the results to the revised Results or Methods section as follows:

<CNN training set vs test set>

(Page 4, Line 29) According to the receiver operating characteristic (ROC) curves, the area under the curve (AUC) for the training data was 0.93 for HLA-A and 0.94 for HLA-B, respectively. In the test data, the AUC was 0.89 for HLA-A and 0.86 for HLA-B.

<RF training set vs test set>

(Page 19, Line 9) For each cohort, we used the other cohorts of the same tumour type as training data. For example, we trained random forests with Hellman and Rizvi cohorts and tested performance on SMC cohort. The AUCs were 0.81 ~ 0.97 for the training data and 0.76 ~ 0.95 for the test data.

To further rule out the possibility of overfitting, we compared the prediction results for the test data to those of other tools (for CNN) and control training models (for RF) as shown in revised Figure 1B and Figure 3A (attached below).

<Performance comparison between the CNN model and other tools>

<Performance comparison between the RF model and control model>

6. Fig 1B. "70.5% of the test datasets (100/132)" I believe this is incorrectly worded and there are only 33 test datasets, which were evaluated using 4 different comparison models. This needs to be reworded (e.g. "outperforms alternative approaches in X to Y% of the datasets").

We thank the reviewer for pointing this out. In revision, we used the latest version of NetMHCpan (NetMHCpan 4.0; Jurtz et al., J Immunol 2017) for comparison with our CNN model. We also used the latest versions of other algorithms (e.g., NetMHCcon) for comparison. The comparison results, shown below, are provided in revised Figure 1B.

During the comparison tasks, we detected undesirable variabilities in the performance levels when the size of training data was not sufficiently high. Therefore, only the HLA alleles with the size of training data > 3,000 were used for performance comparison. Each comparison between our CNN and other tools is described in our revised manuscript as follows:

(Page 5, Line 18) In terms of AUC, our method was superior to SMMPMBEC, ANN, NetMHCcon, and NetMHCpan for 100%, 100%, 90%, and 70% of the test cases. With regards to the F1 score, our CNN was superior to all the methods for 80% of the test cases.

7. The application of 5-fold cross-validation in training the RF model is not adequately explained. Was this used to set the hyperparameters for training on the full dataset? Was it somehow used to create the model that was evaluated on the test set?

Five-fold cross-validation was performed to optimize the hyperparameters for the RF training as asked by the reviewer. We applied 5-fold cross-validation only for the training data independently of the test set. During revision, we substantially increased the amount of cohort data to validate the robustness and consistency of our RF prediction (revised Table 1 attached below). In particular,

we generated our own lung cancer cohort (n = 122). For each cohort, we used the other cohorts of the same tumor type as training data. For example, we trained RF with Hellman and Rizvi cohorts and tested performance on SMC cohort.

Cohort name	Tumor type	Cohort size	Target checkpoint	Reference
SMC	Lung cancer	122	PD-1/PD-L1	This work
Hellmann	Lung cancer	75	PD-1 & CTLA-4	Ref 25
Rizvi	Lung cancer	34	PD-1	Ref 24
Van Allen	Melanoma	110	CTLA-4	Ref 21
Snyder	Melanoma	64	CTLA-4	Ref 22
Roh	Melanoma	56	PD-1 & CTLA-4	Ref 10
Riaz	Melanoma	68	PD-1	Ref 12

8. Were the 269 genes all the genes "that harbour deleterious or damaging mutations" as described in the text, or was this number determined through cross-validation? The number of features (269) is greater than the number of training examples (174). Unless there are a lot of correlated features, this would lead to overfitting if no regularization is performed.

Because we substantially increased the number of cohort samples as described above in our response to comment #7, the number of features in the updated training became smaller than the number of samples. We also tested whether there were correlated features leading to overfitting based on the Spearman's correlation between gene mutations within the training data. As a result, there was no highly correlated features showing rho value over 0.9 in both cohorts. Only 4 genes showed rho value over 0.8 in the lung cancer cohorts.

9. The 100 tests are not explained at all in the results section. What is the size of the 100 subsets chosen? Are they sampled with or without replacement? This sort of analysis gives an estimate of variance, not "predictive performance" as stated in the text.

In our original submission, the number of functional mutations was greater than synonymous mutations. This is why we used the 100 random samplings of functional mutations to match the number of synonymous mutations. However, during revision, we considerably increased the size of training/test data (see our response to comment #7). After the restructuring of the training process, there was no excess of functional mutations as compared to synonymous mutations, thereby removing the need to do the random sampling.

10. "accuracy > 3 for the clinical melanoma data and accuracy > 1 for the TCGA". Note sure what is being communicated here. How is accuracy being measured? Are those suppose to be percentages?

The "accuracy" in our manuscript implies "the mean decrease in accuracy," which is an indicator of how much each feature contributes to RF prediction. The calculation of this value was described in the manuscript as below:

(Page 22, Line 24) The 'variable importance' of each feature (mutated gene) was evaluated on the basis of the mean decrease in accuracy as implemented in the randomForest R package. Specifically, the importance of the k th feature was measured as the degree of decrease in prediction accuracy upon random permutation of all values in the k th feature of the training dataset.

The RF training and testing using TCGA samples showed significantly lower levels of the mean decrease in accuracy than that using the clinical cohort data. Therefore, we lowered the threshold (accuracy > 1) to select important features from the TCGA model.

11. Fig. 4. There are quite a few problems here. "Of the 46 non-responders, our CNN method predicted a low load of neoantigens for 15 samples (grey bars in the upper graph of Fig. 4A) and a high load of neoantigens for 31 samples (blue and orange bars in the upper graph of Fig. 4A)." "(A) Neoantigen load and resistance parameters for 47 nonresponders to checkpoint blockade in Roh et al.'s cohort" 46 != 47 and there are actually 52 datapoints in the figure. Roh's dataset has 56 patients. There is a "Responder" and a "Non-responder" label on the figure, but it isn't at all clear what they are trying to communicate and the text of the paper makes no reference to the responders. This is unfortunate, since in order to justify the claim that a method has predictive power, it must be able to distinguish between the two classes of interest. Perhaps the first six bars represent the responders? But each of these examples has a comparable example among the non-responders - there is no clear discrimination visible in this figure.

First, we apologize for this confusion in defining the patient response. We simply used the definition of responders and non-responders described in the corresponding literature. For example, the following are the description of the clinical response for the Roh cohort: “As such, patients were first defined as those having either (i) a complete response [disappearance of all target lesions, reduction in any pathological lymph nodes (whether target or not) in short axis to <10 mm, and the appearance of no new lesions], (ii) a partial response (at least a 30% decrease in the sum of diameters of target lesions, no PD in nontarget lesions, and the appearance of no new lesions), (iii) progressive disease (at least a 20% increase in the sum of diameters of target lesions, taking the smallest sum or baseline as reference, with a minimum absolute increase of 5 mm, and/or the development of any new lesions), or (iii) stable disease [neither sufficient decrease to designate complete response/partial response nor increase to qualify as progressive disease (again using the smallest sum of appropriate diameters as a reference)]. All image responses were vetted with ≥ 2 serial images over a ≥ 6 -month interval between baseline and assignment of response. RECIST 1.1 quantification of response was then used to assign patient designation as responder (complete response, partial response, or stable disease for ≥ 6 months) or nonresponder (progressive disease or stable disease with <6-month duration).

And the following are the definition of the clinical response for our lung cancer cohort, as provided in the Methods section under the subsection titled “SMC cohort for checkpoint blockade in lung cancer”:

(Page 17, Line 20) A total of 122 advanced non-small cell lung carcinoma patients who were treated with anti-PD-1/PD-L1 from 2014 to 2017 at Samsung Medical Center were enrolled for this study. The clinical response was evaluated by the Response Evaluation Criteria in Solid Tumours (RECIST) version 1.1 with a minimum 6-month follow-up. The response to immunotherapy was classified into durable clinical benefit (DCB, responder) or non-durable benefit (NDB, non-responder). Partial response (PR) or stable disease (SD) that lasted more than 6 months was considered as DCB/responder. Progressive disease (PD) or SD that

lasted less than 6 months was considered as NDB/non-responder. Progression-free survival (PFS) was calculated from the start date of therapy to the date of progression or death, whichever is earlier. Patients were censored at the date of the last follow-up for PFS if they were not progressed and alive. We complied with all relevant ethical regulations for work with human participants. Informed consent was obtained. This study was approved from the institutional review board at Samsung Medical Center (2018-03-130 and 2013-10-112).

The first six bars corresponded to responders as assumed by the reviewer. In the original figure, the bars indicated neoantigen load while divided by whether they belonged to a responder or non-responder. Our RF predictor classified non-responders despite having high neoantigen load as colored bars (except gray bars) in the original figure. Also, 31 samples did not have high neoantigen load whereas 32 samples had high neoantigen load. We profiled only 52 samples of Roh cohort although the original sample size was 56 because the 4 samples lacked clinical response information.

During revision, we performed more extensive and systematic analyses by including our own cohort and increasing cohort data. We removed the original figure and provided the systematic prediction results in revised Figure 3A as shown below.

12. "Two bar plots below depict neoantigen load estimated by the CNN model and NeMHCpan." There is only one bar plot (CNN) in 4A.

This statement was about the previous example figure that we removed to provide more systematic and extensive results with extended cohort data (see our response comment #11).

13. "The grey bars mark the cases in which no therapeutic response is supported by low neoantigen load." Why are some grey bars above the drawn threshold line then?

This statement was also about the previous example figure that we removed. In the original figure, the responders having high neoantigen load were colored in grey. Like non-responders having low neoantigen load, the response of these patients could be explained by neoantigen load. We referred to this as "Response/non-response supported by neoantigen level" as seen in the original figure.

14. What is a mutation signature? It isn't defined. If it is the score produced by the RF, that isn't a "signature," that's a single number. What threshold is used to draw a green box in Fig 4A?

We apologize for the lack of clarity in the description of “mutation signature”. This is a prediction score (a single number) produced by the RF predictor as pointed out by the reviewer. The scores over 0.5 were defined as positive calls and indicated as the green boxes in original Figure 4A. However, this figure has been removed. Now the term, “mutation signature,” has been substituted by “exomic prediction score” in revised manuscript as highlighted in red.

15. "from simple logistic regression, the mutation signatures only were capable of predicting resistance at high accuracy when our CNN model was used for neoantigen identification (Fig 4B)" "AUC (upper) and variable importance (lower) for the regression of resistance on the mutation signatures when resistance was defined by neoantigen load estimated by CNN or NetMHCpan." This is confusing and misleading to the point I'm not sure what 4B,C,D are showing. Based on the results section, this should be showing that "profiling functional mutations together with neoantigens accurately predicts therapeutic response," but based on the caption text, it is showing that the functional mutations (and other indicators) can be fit in a linear model to the predictions of neoantigen level, which is not the same. I believe the caption is correct (since the neoantigen score is not shown with a weight in the heatmaps) which means, unfortunately, there is no combining of the two approaches presented. Equating "resistance" with the neoantigen level seems wrong, especially considering Fig 4A. Some explanation of why this exercise is informative/useful would be appreciated.

We apologize for the lack of clarity causing confusion. When we said "profiling functional mutations together with neoantigens accurately predicts therapeutic response," we meant that neoantigen load estimated by the CNN contributed to accurate patient stratification and thus the proper construction of training data for resistance prediction based on functional mutation profiling by the RF classifier. We created the following cartoon for better understanding of our approach (Supplementary Fig. 7).

We use the term “resistance” because patients with high neoantigen load are expected to respond to checkpoint blockade but may bear resistance because of functional mutations that promote immune evasion. The final prediction score by the RF, indicating whether the given patient would bear therapeutic resistance, was referred to as the “exomic prediction score” because exome data were used for prediction by the RF.

In original Figure 4A, we used the exomic prediction score alone to test its explanatory power for the resistance through the linear regression analysis. Additional known resistance factors such as copy number alteration and heterogeneity were used together with the exomic prediction score to measure combined explanatory power for the resistance through the lasso, elastic net, and ridge regression analysis as described in original Figure 4B, C, and D. The example table below illustrates the input for the regression analyses.

Patient ID	Exomic prediction score	Immun-related pathway mutation	Copy number gain	Heterogeneity	...	Resistance
Patient1	0.4	0	0	0.2	1	0

Patient2	0	1	1	0.1	1	1
Patient3	0.2	0	0	0	1	1
Patient4	0.1	1	0	0.3	0	0
Patient5	0.9	0	1	0	0	1
Patient6	0	0	1	0.7	0	1
...	0	1	0	0	1	0

In our original submission, we applied this only for a melanoma cohort (Roh) because the additional resistance factors were available. During revision, we generated such data for our own (SMC) cohort, enabling us to reproduce the same analysis in lung cancer. The results are provided in revised Figure 4 as attached below.

One of the points that we want to emphasize from the results is that resistance prediction by the exomic prediction score performs better when our CNN was used than NetMHCpan was used for neoantigen load estimation, thereby highlighting the importance of the combined approach.

16. The selection of y-axis scales in 4B,C,D is inconsistent and misleading.

We have now corrected the y-axis scales to be consistent within each analysis as seen below.

17. An AUC doesn't make much sense for a regression model - it is a measure of classification accuracy. Why would this be used?

In the regression analysis, we tested how well our RF prediction and other known features predicted therapeutic resistance. We therefore had to measure the AUC as the accuracy of classification.

18. What is needed is evidence the two approaches can be combined to predict therapeutic response.

Please see our response to comment #15. First, we observed that neoantigen load estimated by our CNN method, which was superior to other methods, could not fully explain or predict the clinical response. This may be because there is a myriad of factors affecting immune responses. We profiled functional mutations and used them as features for RF training to predict non-responders despite high neoantigen load. Resistance prediction (by the exomic prediction score) showed considerable performance consistently across different cancer types when combined with CNN neoantigen prediction (revised Figure 4).

Reviewers' comments:

Reviewer #1 (Remarks to the Author):

General comments

In the revised manuscript Kim and colleagues made an effort to address my previous concerns and added new results. Specifically, another cohort (the Samsung Medical Center) with 122 patients treated with antiPD1/PDL1 antibodies was included, which is major improvement of the manuscript. Additionally, analysis with latest prediction tools as well as TCR tools as suggested was included. The manuscript and the figures were appropriately and extensively redone. While the manuscript is now more convincing, there are still some issues that need to be addressed as shown below.

Specific comments

1. The authors should include also the results of additional predictors (neoantigen load, and wherever possible expression of specific genes including PD1, PD-L1, and cytotoxic activity as reported in Rooney et al, Cell, 160(2015), 48-61) in Figure 3A in order to have comparative analysis.
2. HLA typing was performed with HLAMiner for the SMC cohort and Seq2HLA for the TCGA data. What was the reason for using different typing methods (TCGA data has also exome-seq data) and how would this fact impact the results?

Reviewer #2 (Remarks to the Author):

The revised manuscript has improved the presentation and added much needed technical details. The authors have also added analysis using additional public data sets that have become available in the interim. However a few technical concerns still remain.

- 1.) Regarding Figure 1 and comparisons of AUC – can the authors quantify if these differences are statistically significant or just numerically higher
- 2.) The authors specify they used a cutoff of neo-antigen count to 70 consistently across all cohorts – can the authors demonstrate where this cut-off originated from? i.e. was the cutoff determined as the optimal cut-off using data from all cohorts – or was this cut off identified in one cohort and then applied to others.
- 3.) Figure 2D the results are not statistically significant – the text should be re-written to state this clearly.
- 4.) The analysis on positive selection in Figure 3D needs significantly more detail to be able to adequately interpret.
- 5.) The results regarding alterations in the EGFR pathway mediating resistance need to be clearly separated from the fact that EGFR mutant tumors typically have lower TMB. A couple of control graphs (genes involved in EGFR pathway resistance and relationship to TMB) would help facilitate this.

Reviewer #3 (Remarks to the Author):

The authors were responsive to the reviewers' feedback and significantly improved the article. They have included a new dataset and clarified their claims surrounding the combined application of their algorithms.

Reviewer #1 (Remarks to the Author):

General comments

In the revised manuscript Kim and colleagues made an effort to address my previous concerns and added new results. Specifically, another cohort (the Samsung Medical Center) with 122 patients treated with antiPD1/PDL1 antibodies was included, which is major improvement of the manuscript. Additionally, analysis with latest prediction tools as well as TCR tools as suggested was included. The manuscript and the figures were appropriately and extensively redone. While the manuscript is now more convincing, there are still some issues that need to be addressed as shown below.

We thank the reviewer very much for the constructive comments and acknowledging the efforts and improvements we made for the manuscript. We provide responses to the additional issues raised by the reviewer below.

Specific comments

1. The authors should include also the results of additional predictors (neoantigen load, and wherever possible expression of specific genes including PD1, PD-L1, and cytolytic activity as reported in Rooney et al, Cell, 160(2015), 48-61) in Figure 3A in order to have comparative analysis.

Neoantigen load, expression levels of PD1, PD-L1, MHC, etc., and cytolytic activity are examples of 'positive contributors' to the clinical benefit of checkpoint blockade. In Figure 3A, we assessed the performance of our models based on functional mutations as 'negative contributors' or resistance factors. For comparative analyses suggested by the referee, we compared our mutation-based prediction (exomic prediction) with other resistance factors, including immune-related pathway mutations, antigen-presenting pathway mutations, copy number alterations (aneuploidy), tumour heterogeneity, etc. in Figure 4.

In Figure 4A and D, we used our prediction score (exomic prediction score) alone to test its explanatory power for therapeutic resistance through the linear regression analysis. Other resistance factors were used together with the exomic prediction score to measure combined explanatory power for the resistance through the lasso, elastic net, and ridge regression analysis as described in

Figure 4B, C, E, and F. The example table below illustrates the input for the regression analyses.

Patient ID	Exomic prediction score	Immun-related pathway mutation	Copy number gain	Heterogeneity	...	Resistance
Patient1	0.4	0	0	0.2	1	0
Patient2	0	1	1	0.1	1	1
Patient3	0.2	0	0	0	1	1
Patient4	0.1	1	0	0.3	0	0
Patient5	0.9	0	1	0	0	1
Patient6	0	0	1	0.7	0	1
...	0	1	0	0	1	0

Also, there were no available transcriptome data for the analyzed cohorts so that it was difficult to use gene expressions as predictors. Cytolytic activity (Rooney et al.) was considered in our analysis of the correlation between neoantigen load and immune activity in the TCGA samples (Fig. 2C). We calculated immune score as the geometric mean of the expression level of not only cytolytic activity but also other parameters (IFN- γ pathway genes, chemokines, and adhesion molecules) as previously described (ref 10).

We created the following cartoon for better understanding of our approach and how resistance was defined (Supplementary Fig. 7).

2. HLA typing was performed with HLAmimer for the SMC cohort and Seq2HLA for the TCGA data. What was the reason for using different typing methods (TCGA data has also exome-seq data) and how would this fact impact the results?

We used Seq2HLA for the TCGA samples because only the RNA-seq data were available at the time of analysis while obtaining raw exome data requires additional approval procedures. In fact, the other 6 cohorts used different HLA typing tools. We were only able to obtain the previous typing results from the cohort publications. HLA typing is quite straightforward, and it is hard to expect remarkably different results from different tools. Moreover, we only used the total count of called neoantigens, which may not vary considerably among different HLA typing tools. As a matter of fact, all the patterns we were able to obtain in this work were consistent among different cohorts and also for the TCGA data even when different HLA typing tools were used. This may support the robustness of our findings.

Reviewer #2 (Remarks to the Author):

General comments

The revised manuscript has improved the presentation and added much needed technical details. The authors have also added analysis using additional public data sets that have become available in the interim. However, a few technical concerns still remain.

We thank the reviewer very much for the constructive and valuable comments.

We feel that our manuscript has been improved significantly. We provide responses to the additional issues raised by the reviewer below.

Specific comments

1. Regarding Figure 1 and comparisons of AUC – can the authors quantify if these differences are statistically significant or just numerically higher

We simply compared the level of AUC numerically. According to the reviewer's suggestion, we attempted to compute standard deviation for each AUC measure. However, because of the small sizes of the test data, we failed to obtain reasonable levels of standard deviations. As stated in our previous revision, we detected undesirable variabilities in performance levels when the size of training data was not sufficiently high. Therefore, only the HLA alleles with the size of training data > 3,000 were used for performance comparison. Despite the relatively large size of the training data, the size of the testing data did not allow us to calculate performance variabilities within each dataset.

Instead, we quantified the differences 'across' the test datasets by counting the number of the test datasets in which our CNN method outperformed the others.

We described this numerical quantification in our manuscript as follows:

(Page 5, Line 17) In terms of AUC, our method was superior to SMMPMBEC, ANN, NetMHCcon, and NetMHCpan for 100%, 100%, 90%, and 70% of the test cases.

With regards to the F1 score, our CNN was superior to all the methods for 80% of the test cases.

2. The authors specify they used a cutoff of neo-antigen count to 70 consistently across all cohorts – can the authors demonstrate where this cut-off originated from? i.e. was the cutoff determined as the optimal cut-off using data from all cohorts – or was this cut off identified in one cohort and then applied to others.

We adopted the cutoff from what was used in McGranahan, N. et al. (ref. 9) as described follows:

(Page 18, Line 25) For each of all cohorts used in this study (Table 1), resistant samples were defined as having > 70 predicted neoantigens as previously suggested⁹ while no clinical benefit was reported from each respective study. All remaining samples were defined as non-resistant samples and trained along with the resistant samples.

Neoantigen load either calculated by NetMHCpan or CNN showed reasonable intercohort variations, thereby enabling us to set a unified threshold.

3. Figure 2D the results are not statistically significant – the text should be re-written to state this clearly.

We rewrote the statement about the results of Figure 2D to reflect the issue as follows:

(Page 6, Line 23) For these samples, our estimate of neoantigen load (left of Fig. 2D) than mutation burden (right of Fig. 2D) showed a better correlation with patient survival. However, this should be interpreted with caution given no statistically significance, which may be attributed to the fact that we simply compared patient survival but not in the setting of checkpoint therapy.

4. The analysis on positive selection in Figure 3D needs significantly more detail to be able to adequately interpret.

We apologize for the lack of details on our positive selection analyses. We revised our manuscript to provide more details as below:

(Page 7, Line 31) We then examined whether the predictive genes identified from our classifier carry the signatures of positive selection. Recent studies

investigated selection patterns at the gene level based on the ratio of nonsynonymous to synonymous mutations across a large number of tumour samples^{32,33}. Positive selection on mutations will lead to the excess of nonsynonymous mutations for given background mutation rates estimated by the frequency of synonymous mutations. In other words, a gene under positive selection will carry an extra complement of driver mutations in addition to passenger mutations. We employed the scores that were previously calculated for each gene by the Bayesian inference³² and statistical model for covariates (*dNdScv*)³³ based on the mutation patterns observed in the TCGA data. Using these scores, we compared the degree of positive selection on the predictive genes with that on genes categorized as antigen-presentation or immune-related pathway. The score distributions for all genes were also considered. As a result, significantly higher positive selection scores, in particular those from the Bayesian inference³², were observed for the predictive genes than for the other groups of genes (Fig. 3D), indicating that our prediction model was based on functional mutations that are subject to positive selection because of their contribution to immune evasion.

5. The results regarding alterations in the EGFR pathway mediating resistance need to be clearly separated from the fact that EGFR mutant tumors typically have lower TMB. A couple of control graphs (genes involved in EGFR pathway resistance and relationship to TMB) would help facilitate this.

We thank the reviewer for this constructive comment. There were few samples that carried specific mutations on the EGFR gene itself in the examined cohorts. Therefore, we considered mutations on all genes in the EGFR pathway. When we first examined the correlation between TMB and the mutation status of the EGFR pathway using the cohort samples, it turned out that the EGFR pathway-mutant tumours had higher TMB than the wildtype tumours as shown in the graph below.

We selected the neoantigen-high group as shown in the following schematic. Therefore, many EGFR pathway mutants must have been included in the neoantigen-high group. According to our prediction models, the EGFR pathway mutants must have a tendency toward therapeutic resistance in the setting of checkpoint inhibitor therapy.

Reviewer #3 (Remarks to the Author):

The authors were responsive to the reviewers' feedback and significantly improved the article. They have included a new dataset and clarified their claims surrounding the combined application of their algorithms.

We thank the referee for reviewing our manuscript and providing constructive and valuable comments. We feel that the comments of the reviewer made our manuscript improve significantly.

REVIEWERS' COMMENTS:

Reviewer #1 (Remarks to the Author):

The authors addressed the issues raised in the review following the revised manuscript satisfactorily.

Reviewer #2 (Remarks to the Author):

the authors have adequately addressed my remaining concerns.